# Pathways for balancing $CO_2$ emissions and sinks

Brian Walsh[1], Philippe Ciais[2], Ivan A. Janssens[3], Josep Peñuelas[4,5], Keywan Riahi[1], Felicjan Rydzak[1], Detlef P. van Vuuren[6,7] & Michael Obersteiner[1]

In December 2015 in Paris, leaders committed to achieve global, net decarbonization of human activities before 2100. This achievement would halt and even reverse anthropogenic climate change through the net removal of carbon from the atmosphere. However, the Paris documents contain few specific prescriptions for emissions mitigation, leaving various countries to pursue their own agendas. In this analysis, we project energy and land-use emissions mitigation pathways through 2100, subject to best-available parameterization of carbon-climate feedbacks and interdependencies. We find that, barring unforeseen and transformative technological advancement, anthropogenic emissions need to peak within the next 10 years, to maintain realistic pathways to meeting the COP21 emissions and warming targets. Fossil fuel consumption will probably need to be reduced below a quarter of primary energy supply by 2100 and the allowable consumption rate drops even further if negative emissions technologies remain technologically or economically unfeasible at the global scale.

[1] International Institute for Applied Systems Analysis, Schlossplatz 1, A-2361 Laxenburg, Austria. [2] Laboratoire des Sciences du Climat et de L'Environnement, CEA-CNRS-UVSQ, 91190 Saint-Aubin, France. [3] University of Antwerp, 2610 Wilrijk, Belgium. [4] CSIC, Global Ecology Unit CREAF-CSIC-UAB, Cerdanyola del Valles 08193, (Catalonia), Spain. [5] CREAF, Cerdanyola del Valles 08193 (Catalonia), Spain. [6] PBL Netherlands Environmental Assessment Agency, 2594 AV The Hague, The Netherlands. [7] Copernicus Institute for Sustainable Development, Utrecht University, 3584 CS Utrecht, The Netherlands. Correspondence and requests for materials should be addressed to B.W. (email: walsh@iiasa.ac.at).

At the Conference of Parties in Paris (COP21), in December 2015, negotiators from 195 countries agreed to 'pursue efforts to limit the (global average) temperature increase to 1.5 °C above pre-industrial levels, recognizing that this would significantly reduce the risks and impacts of climate change'. The text of the Paris Agreement further specifies 'Parties aim to reach global peaking of greenhouse gas emissions as soon as possible...and to undertake rapid reductions thereafter in accordance with best available science, so as to achieve a balance between anthropogenic emissions by sources and removals by sinks of greenhouse gases in the second half of this century'[1]. With these parallel goals, the agreement requires complete decarbonization of both the energy and land use, land-use change and forestry (LULUCF) sectors before the end of the century, but the pace of this transition is left to nationally determined contributions in accordance with 'the best available science'.

Integrated assessment models can be used to link the emissions and climate targets of the Paris Agreement to necessary transitions in the energy and LULUCF sectors[2]. The continuation of recent trends in global land, energy and carbon systems defines a baseline scenario, or business-as-usual (BAU). Disregarding the possibility of transformative policy and technological shifts, BAU projects global carbon emissions, atmospheric carbon concentration ($C_A$) and average surface temperature relative to preindustrial ($\Delta T$) through the end of the century. Alternative emissions pathways, differentiated from the baseline in terms of the development of the energy or LULUCF sectors, can be used together with BAU to define a probable carbon budget for the achievement of the COP targets.

In this analysis, we use the FeliX model to derive emissions, $C_A$, and $\Delta T$ projections for a Fossil Fuels scenario, in which the primary energy market share of fossil fuels remains near constant through 2100 and for two scenarios in which reliance on renewable energies (RE) accelerates modestly (RE-Low) and rapidly (RE-High) relative to the baseline. Together with BAU, these scenarios are defined by their respective primary energy profiles. Finally, we examine the potential impact of additional emissions mitigation in the RE-Low and RE-High scenarios through carbon capture and sequestration (CCS) or utilization.

We find that, barring unforeseen and transformative technological advancement, anthropogenic emissions need to peak within the next 10 years, to maintain realistic pathways to meeting the COP21 emissions and warming targets. Fossil fuel consumption will probably need to be reduced below a quarter of primary energy supply by 2100 and the allowable consumption rate drops even further if negative emissions technologies remain technologically or economically unfeasible at the global scale.

## Results

**Atmospheric flux ratio**. The concept of a carbon budget involves multiple dynamic, interrelated components of the global carbon cycle and can be defined in a number of ways. As a figure of merit, we define an atmospheric flux ratio ($R_{AF}$) as the ratio of net $CO_2$ emissions (anthropogenic sources minus artificial sinks) to net $CO_2$ uptake by natural sinks (that is, plant, soil and ocean systems).

The atmospheric flux ratio characterizes annual changes in the atmospheric carbon burden. Atmospheric flux ratios greater than unity ($R_{AF} > 1$) indicate increasing atmospheric carbon concentrations, associated radiative forcing and temperatures. Ratio values between zero and unity ($0 < R_{AF} < 1$) indicate net negative atmospheric carbon flux due to net ocean and land sink uptake, an important milestone on the path to climate stabilization. Finally, values below zero ($R_{AF} < 0$) indicate net negative anthropogenic emissions—that is, the achievement of the COP carbon emissions target. We calculate recent historical values of $R_{AF}$ on the basis of data from the Intergovernmental Panel on Climate Change (IPCC)[3] and use the FeliX model to project $R_{AF}$ through 2100 for all scenarios.

Based on the most recent available IPCC data for global atmospheric carbon flux, we calculate $R_{AF} = 1.9 \pm 0.2$ for the period 2002–2011 (cf. Fig. 1). This value is in good agreement with FeliX model results, which estimate $R_{AF} = 2.1 \pm 0.2$, indicating that net anthropogenic carbon emissions are roughly double the combined net uptake by plants, soil and oceans. Looking in more detail at year 2015 of the FeliX model, primary energy consumption totals nearly 600 EJ per year (cf. Fig. 2) and net emissions from the energy and LULUCF sectors total 10.4 and

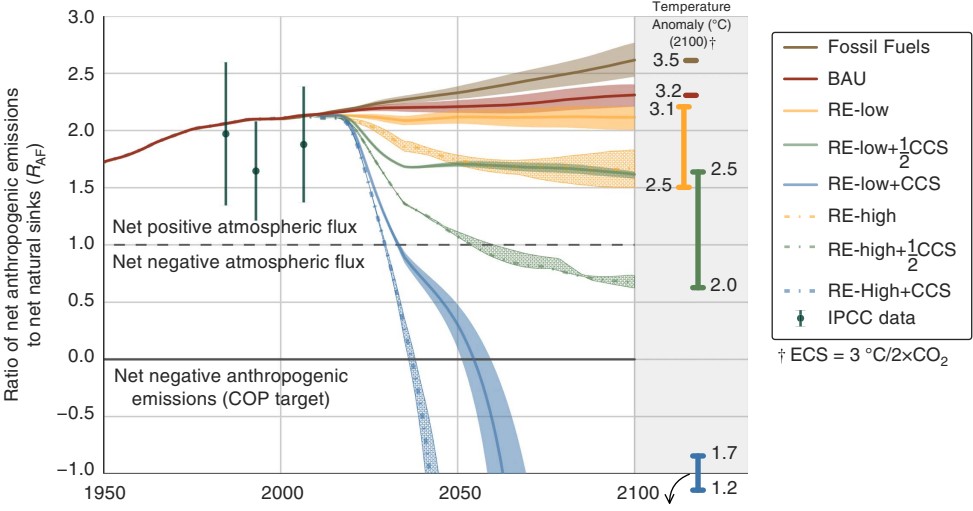

**Figure 1 | Atmospheric carbon flux ratios.** Ratios are calculated annually as the ratio of net anthropogenic carbon emissions (energy and land-use emissions minus anthropogenic sinks) to net carbon sequestration by global plant, soil and ocean systems. Shaded ranges around the central value of each scenario indicate sensitivity of results to primary energy demand. Global surface temperature anomalies projections ($\Delta T$) in 2,100 are indicated at the right, where each coloured bar treats the RE-Low and RE-High scenarios as the endpoints of a continuous range of energy sector decarbonization, plus a fixed rate of CCS. Historical values of $R_{AF}$ and associated errors (cf. equation (12)) from the IPCC are indicated by green bars[3].

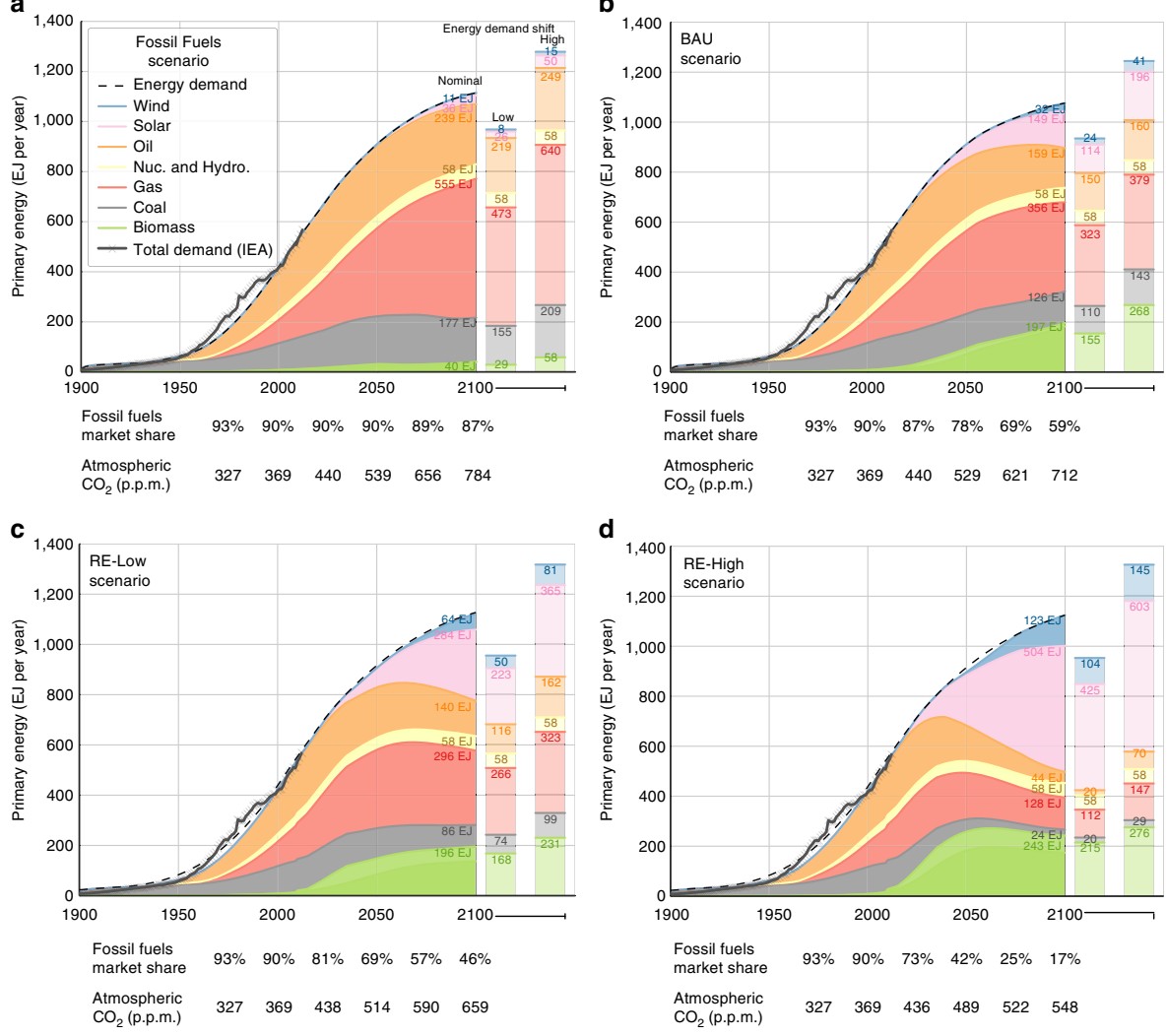

**Figure 2 | Time series of global primary energy consumption.** Nominal demand in the (**a**) Fossil Fuels, (**b**) BAU, (**c**) RE-Low and (**d**) RE-High scenarios is shown as time series in each panel. Energy profiles in year 2100 of alternative pathways, used to determine model sensitivity to primary energy demand, are displayed as columns at right. At bottom, time series of fossil fuel market share (as a percentage of total energy consumption) and atmospheric $CO_2$ concentration are displayed at quarter-century intervals. Source of historical data on total primary energy demand: International Energy Agency[39].

1.0 PgC, respectively (*cf*. Figs 3a and 4a). In the same year, the carbon content of the atmosphere increases by 6.3 PgC and the ocean and land sinks each take up 2.7 PgC (*cf*. Fig. 5).

Carbon uptake by natural sinks is dependent on anthropogenic emissions via climactic and chemical feedback mechanisms[4,5]. In decarbonization scenarios, net carbon uptake by the land and ocean systems approaches zero as anthropogenic emissions also decrease to zero and this dynamic relationship makes a moving target of net negative atmospheric carbon flux. In this section, we present FeliX scenario results and use $R_{AF}$ to evaluate energy and LULUCF transition pathways relative to the COP targets.

**The BAU scenario.** The baseline scenario projects an increase in anthropogenic emissions an additional 20% beyond the absorptive capacity of natural sinks by 2100 ($R_{AF} = 2.3 \pm 0.6$) (*cf*. Fig. 1). In the nominal energy pathway, REs (that is, solar, wind and biomass) grow at an average annual rate of 4.0% from 2013 through 2100, causing a decline in the total market share of fossil fuels from 90% (2015) to 60% (2100) of primary energy

supply. In absolute terms, however, consumption of conventional fuels grows until 2060 to meet growing primary energy demand (*cf*. Fig. 2b). Fossil fuel emissions peak at 12.7–14.5 PgC per year around 2055, then decrease to 10.9–12.6 PgC per year by 2100. Emissions from REs increase from 0.1 PgC per year in 2015 to 0.5–0.7 PgC per year in 2100, reflecting the costs of bioenergy harvesting, transportation and processing (*cf*. Fig. 3).

In the LULUCF sector, emissions mitigation from the continuation of recent progress combating deforestation is completely offset by emissions from anticipated conversion of natural forests to managed forests and plantations for bioenergy production[6,7]. Net LULUCF emissions decrease to 0.4–1.3 PgC per year in 2100 (*cf*. Fig. 4).

Overall, annual anthropogenic emissions in the BAU scenario peak at 13.5–15.6 PgC per year around 2054, then decline modestly over the second half of the century (*cf*. Fig. 5a). Annual ocean $CO_2$ flux increases to a peak of 3.2–3.4 PgC per year around 2065, after which point ocean flux stabilizes as the chemical and temperature feedbacks cancel each other out. Net carbon flux into the land sink peaks at the same

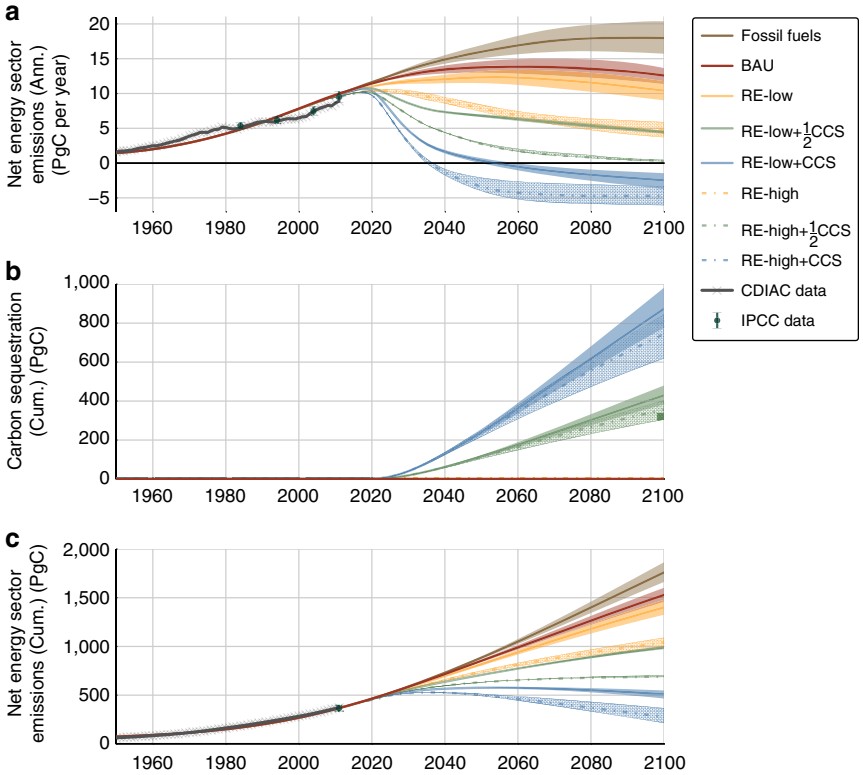

**Figure 3 | Energy sector carbon emissions.** (**a**) Annual net emissions (PgC per year); (**b**) cumulative emissions mitigation from CCS (PgC); (**c**) cumulative net emissions (PgC). Sources of historical data in **a,c**: IPCC AR5 (green bars)[3] and CDIAC (grey series)[33].

magnitude (3.1–3.4 PgC per year) around 2046 and declines to 2.3–2.9 by 2100 due to the temperature feedback (*cf.* Fig. 5).

Relative to the twentieth century, cumulative anthropogenic emissions increase 225–246% (1,250–1,419 PgC) in the present century (*cf.* Fig. 6b). The BAU energy emissions pathway leads to $C_A = 684$–743 p.p.m. and $\Delta T = 3.1$–3.3 °C in 2100. To contextualize these endogenous results using IPCC benchmarks, we note that the energy and LULUCF sectors, as modelled in the BAU scenario, generate cumulative carbon emissions and temperature anomalies similar to Representative Concentration Pathway (RCP) 6.0, one of the IPCC's four benchmark concentration pathways for $CO_2$ and other greenhouse gases[6,7].

**The Fossil Fuels scenario.** The Fossil Fuels scenario projects an increase in anthropogenic emissions to at least 2.5 times the absorptive capacity of natural sinks by 2100 ($R_{AF} = 2.6 \pm 0.7$) (*cf.* Fig. 1). In the nominal energy pathway, REs grow at an average annual rate of 2.2% from 2013 through 2100. The primary energy market share of fossil fuels remains steady near 90% over the course of the century, whereas absolute fossil fuel consumption increases to 847–1097 EJ per year by 2100 (*cf.* Fig. 2a). Fossil fuel emissions peak at 16.0–20.1 PgC per year close to the end of the century, whereas emissions from REs do not exceed 0.2 PgC per year (*cf.* Fig. 3). LULUCF emissions fall relative to the baseline due to the eventual depletion of easily deforested areas and because diminished bioenergy demand reduces conversion of forests to managed forests and plantations (*cf.* Fig. 4).

Annual anthropogenic emissions in the Fossil Fuels scenario peak at 16.0–20.7 PgC per year around 2100. Net $CO_2$ flux into the ocean peaks at 3.5–3.8 PgC per year around 2088 and net carbon flux into the land sink peaks at 3.4–3.9 PgC per year

around 2065 (*cf.* Fig. 5). Cumulative anthropogenic emissions reach 1,435–1,642 PgC (a 250–300% increase over twentieth century emissions) $C_A = 749$–823 p.p.m. and $\Delta T = 3.4$–3.6 °C in 2100 (*cf.* Fig. 6a–d).

**The RE-Low scenario.** The RE-Low scenario, constructed along the lines of moderate climate action scenarios[8], projects persistent anthropogenic emissions at least two times larger than the absorptive capacity of natural sinks by 2100 ($R_{AF} = 2.1 \pm 0.5$) (*cf.* Fig. 1). In the nominal energy pathway, REs grow at an annual rate of 4.4% from 2013 through 2100, driving the market share of conventional fuels below half of primary energy supply in 2100.

Although absolute fossil fuel consumption does not begin to decline until after midcentury, it peaks lower (590–688 EJ per year around 2054) and declines more rapidly (to 455–583 EJ per year in 2100) than in the baseline (*cf.* Fig. 2c). Annual fossil fuel emissions peak in 2050 at 11.1–12.8 PgC per year and decrease to 8.3–10.7 PgC per year in 2100, and emissions from REs grow to 0.7–1.1 PgC per year in 2100 (*cf.* Fig. 3).

Assuming a global, concerted effort to achieve the COP targets, policies driving decarbonization of the energy sector in the RE-Low scenario are linked by construction with land-use change restrictions eliminating 'unnecessary' deforestation or forest loss resulting from poverty, conflict, poor governance, lack of knowledge and other sub-optimal uses of land[9]. FeliX calculates unnecessary deforestation endogenously as the difference between fallow agricultural land and cumulative deforestation after 2011. This optimization of land use offsets emissions from forest conversion to managed forests and plantations, causing annual LULUCF emissions to decrease to 0.1–0.3 PgC per year in 2100 (*cf.* Fig. 4a).

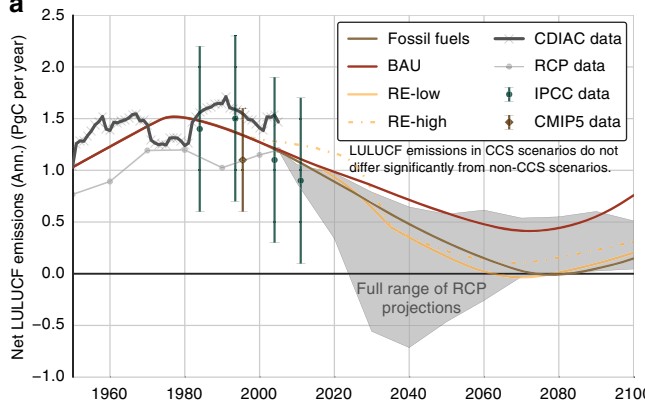

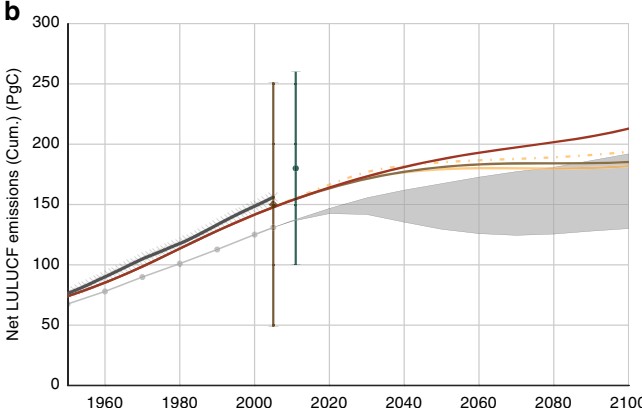

**Figure 4 | LULUCF sector carbon emissions.** (**a**) Annual net emissions (PgC per year); (**b**) cumulative net emissions (PgC). Sources of historical data and errors (*cf.* equation 12): IPCC AR5 (green bars)[3], CMIP5 (brown bars)[5], CDIAC (thick grey lines)[33] and RCP database (thin grey lines)[7]. For comparison, RCP projections through 2100 are indicated as a single grey range.

In total, net anthropogenic emissions peak at 12.2–13.2 PgC per year in 2048 and subsequently decline to 9.3–12.0 PgC per year by 2100 (*cf.* Fig. 5a). In response, cumulative ocean and land sink carbon fluxes over the same period decrease by 7% and 10%, respectively, relative to BAU. These pathways generate a 13% reduction in cumulative emissions relative to the baseline for the period 2011–2100. Although they represent progress in terms of atmospheric carbon ($C_A = 633 − 686$ p.p.m.) and surface temperature anomaly ($\Delta T = 2.9 − 3.1\,°C$), the energy and LULUCF transitions in the RE-Low scenario fall short of global emissions and warming targets by 2100 (*cf.* Fig. 6).

**The RE-High scenario.** The RE-High scenario describes an accelerated transition to REs in line with aggressive climate action scenarios[8]. In this pathway, anthropogenic emissions remain at least 50% higher than natural sink flux ($R_{AF} = 1.7 ± 0.6$), but this ratio belies significant progress towards the COP targets.

From 2013 to 2100, REs grow at 5.0% annually, causing absolute fossil fuel consumption to peak as soon as 2022 at 523–532 EJ per year, then decline to less than a fifth of primary energy supply (166–233 EJ per year) by 2100 (*cf.* Fig. 2). Energy sector emissions peak at 10.3–10.5 PgC per year in 2023 and fall to 4.0–5.7 PgC per year in 2100 (*cf.* Fig. 3). With the elimination of unnecessary deforestation, net LULUCF emissions fall to 0.2–0.4 PgC per year in 2100 or slightly greater than the

corresponding RE-Low value due to land-use change in support of expanded bioenergy production (*cf.* Fig. 4).

Summing these effects, net anthropogenic emissions peak at 11.5–11.7 PgC per year in 2022 then decrease to 4.2–6.1 PgC per year in 2100. In response, cumulative carbon uptakes by the ocean and land sinks decrease by 23% and 35%, respectively, relative to BAU from 2011 to 2100 (*cf.* Fig. 5). Cumulative emissions decrease by 42% relative to BAU over the same period. At the end of the century, $C_A = 532–563$ p.p.m. and $\Delta T = 2.5–2.6\,°C$ (*cf.* Fig. 6), making the RE-High scenario an approximation of IPCC benchmark RCP 4.5 (refs 6,7).

**CCS or utilization.** Taken together, the Fossil Fuels, BAU, RE-Low and RE-High scenarios define a range of plausible energy-emissions pathways. As we have seen, however, even 5% annual growth in REs sustained through the twenty-first century fails to achieve critical benchmarks for both $R_{AF}$ and $\Delta T$. Global transformations of the energy and LULUCF sectors need to be even more ambitious than in the RE-High scenario to achieve the COP targets, suggesting that additional socio-economic and technological shifts must be considered.

CCS is implemented as a mitigation wedge within the RE-Low and RE-High scenarios to quantify the additional emissions mitigation necessary to achieve the COP targets. As variations on the energy and LULUCF sector transitions already discussed, we model the effects of CCS technologies scaled up after 2020 to 50% ($\frac{1}{2}$CCS) or 100% (CCS) of global energy infrastructures. Assuming an 80% capture efficiency, CCS systems capture and permanently sequester up to 40% of gross energy sector emissions in the RE-Low + $\frac{1}{2}$CCS and RE-High + $\frac{1}{2}$CCS variants and up to 80% of gross energy sector emissions in the RE-Low + CCS and RE-High + CCS variants.

To achieve these capture rates while keeping up with primary energy demand growth, the $\frac{1}{2}$CCS and CCS pathways assume geometric growth (30% and 34% per year, respectively, from 2016 through 2040) in the amount of new carbon sequestered each year. Relative to the actual current value (7.3 MtC per year)[10], annual carbon capture expands by a factor of 630–720 by 2040 in $\frac{1}{2}$CCS scenarios and by a factor of 1,310–1,490 by 2040 in the CCS scenarios. From 2040 through the end of the century, CCS capacity expands an additional 10–40% in both pathways to account for rising marginal costs of additional efficiency improvements and infrastructural expansion (*cf.* Supplementary Fig. 1a).

In the RE-Low scenario with $\frac{1}{2}$ CCS, $R_{AF} = 1.6 ± 0.6$, $C_A = 524–535$ p.p.m. and $\Delta T = 2.4–2.5\,°C$ in 2100. For this energy profile, the $\frac{1}{2}$ CCS infrastructure avoids emissions up to 6.1 PgC per year and reaches a cumulative total of 383–429 PgC in 2100 (*cf.* Fig. 3b). In the RE-High scenario, the $\frac{1}{2}$ CCS infrastructure results in $R_{AF} = 0.7 ± 0.9$, $C_A = 443–449$ p.p.m. and $\Delta T = 1.9–2.0\,°C$ in 2100. The technology sequesters up to 5.3 PgC per year for a cumulative total of 305–403 PgC in 2100.

With CCS applied to the entire energy sector, the RE-Low + CCS scenario achieves $R_{AF} = − 4.1 ± 5.4$, where the outsized error is due to the vanishing denominator, representing natural sinks (*cf.* Fig. 5b,c). CCS infrastructures eliminate up to 12.6 PgC per year, with 776–981 PgC in storage by 2100. At the end of the century, $C_A = 393–405$ p.p.m. and $\Delta T = 1.6–1.7\,°C$, making RE-Low + CCS an approximation of IPCC benchmark RCP 2.6 (refs 6,7).

Finally, $R_{AF}$ rises again to $R_{AF} = − 2.1 ± 1.0$ in RE-High + CCS, as the removal of carbon from the atmosphere reverses the sign of the chemical coupling between the ocean and land reservoirs and the atmosphere, transforming them into net sources of carbon (*cf.* Fig. 5b,c). In this scenario, CCS eliminates

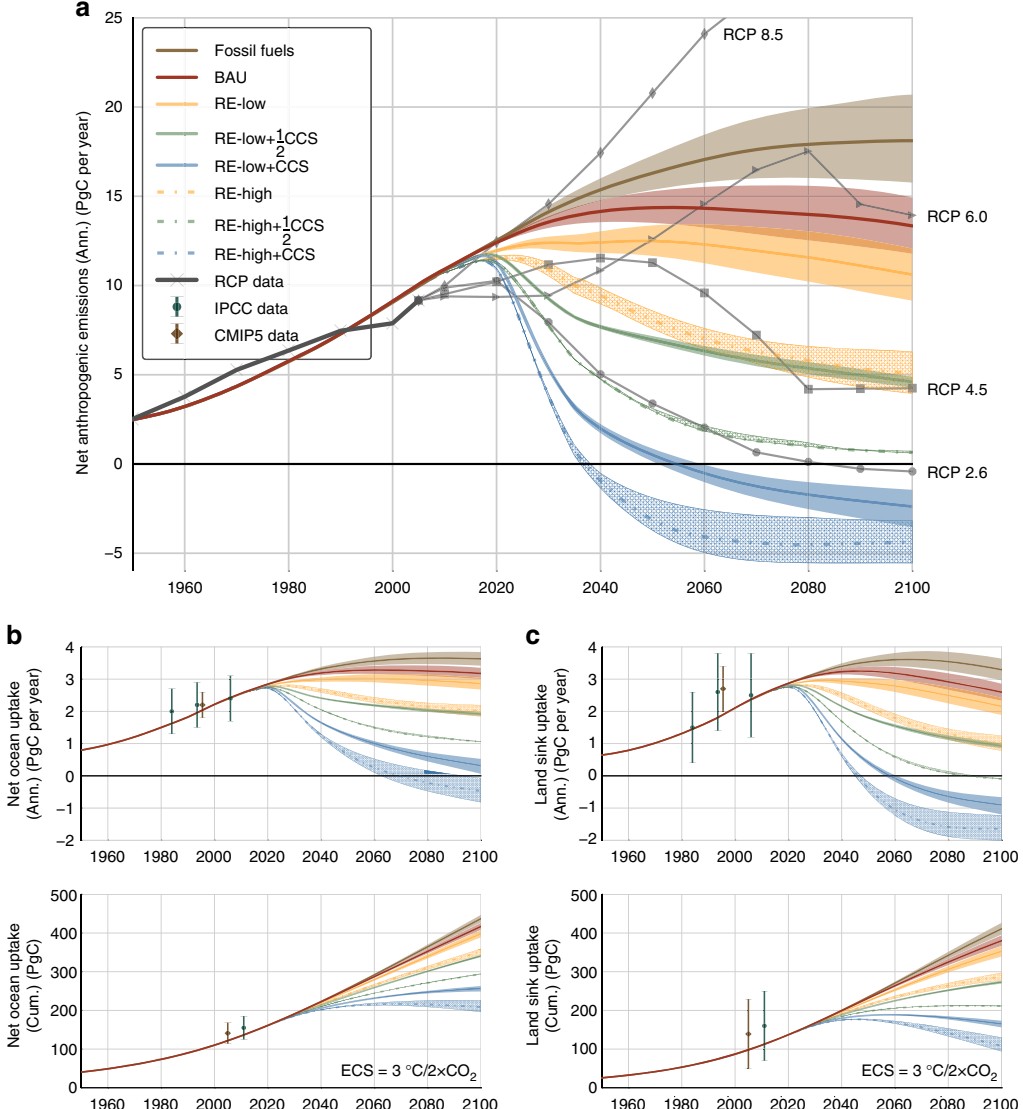

**Figure 5 | Global carbon fluxes.** (**a**) Annual net anthropogenic emissions (PgC per year). RCP data (thick grey line) and projections (thin grey series) shown for comparison; annual and cumulative carbon flux from the atmosphere to the (**b**) oceans and (**c**) land sink (biosphere & pedosphere). Historical data with associated errors (*cf.* equation (12)): IPCC AR5 (green bars)[3] and CMIP5 (brown bars)[5].

up to 11.2 PgC per year and stores a cumulative total of 620–844 PgC by 2100 (*cf.* Fig. 3b). At the end of the century, $C_A = 328–363$ p.p.m. and $\Delta T = 1.2–1.4\,°C$, making this scenario the only one consistent with global warming below 1.5 °C and beginning a return to pre-industrial atmospheric carbon concentration (*cf.* Fig. 6).

**Scenario sensitivity analysis**. Errors on $R_{AF}$ are calculated by propagating uncertainties on each of the component fluxes, as estimated by the IPCC and discussed in the Methods section of this analysis, into the final figure of merit (*cf.* Supplementary Table 1).

Within each scenario, we have reported $C_A$, $\Delta T$ as numerical ranges in the text and as shaded regions in all figures, to indicate the sensitivity of the results to alternative primary energy demand projections. Primary energy demand shifts are the cumulative effect of exogenous $\pm 0.2\%$ annual increments in per capita energy demand after 2015, while market share of each fuel is held near constant (*cf.* Fig. 2). The effect of this shift ranges

from $\Delta C_A \pm 16$ p.p.m. in RE-High (2100) to $\Delta C_A \pm 37$ p.p.m. in Fossil Fuels (2100).

Future population growth is the leading socio-economic source of uncertainty, whereas the initial net primary productivity (NPP), land sink carbon residence time and equilibrium climate sensitivity (ECS; or the global temperature increase resulting from a doubling of the atmospheric carbon load) are the leading biogeophysical sources of uncertainty in emissions and warming pathways[11]. High and low shifts from the nominal value of these and other fundamental model parameters are used to determine the sensitivity of the BAU scenario (*cf.* Supplementary Table 2).

**Discussion**

Before proceeding, we offer a few remarks about $R_{AF}$ and its aptness as a figure of merit. First, we acknowledge that the roles of the ocean and land sinks are often left implicit in carbon budget analyses and the text of the Paris Agreement itself leaves ambiguous the definition of 'net zero emissions' insofar as it does not specify whether sinks need be directly anthropogenic

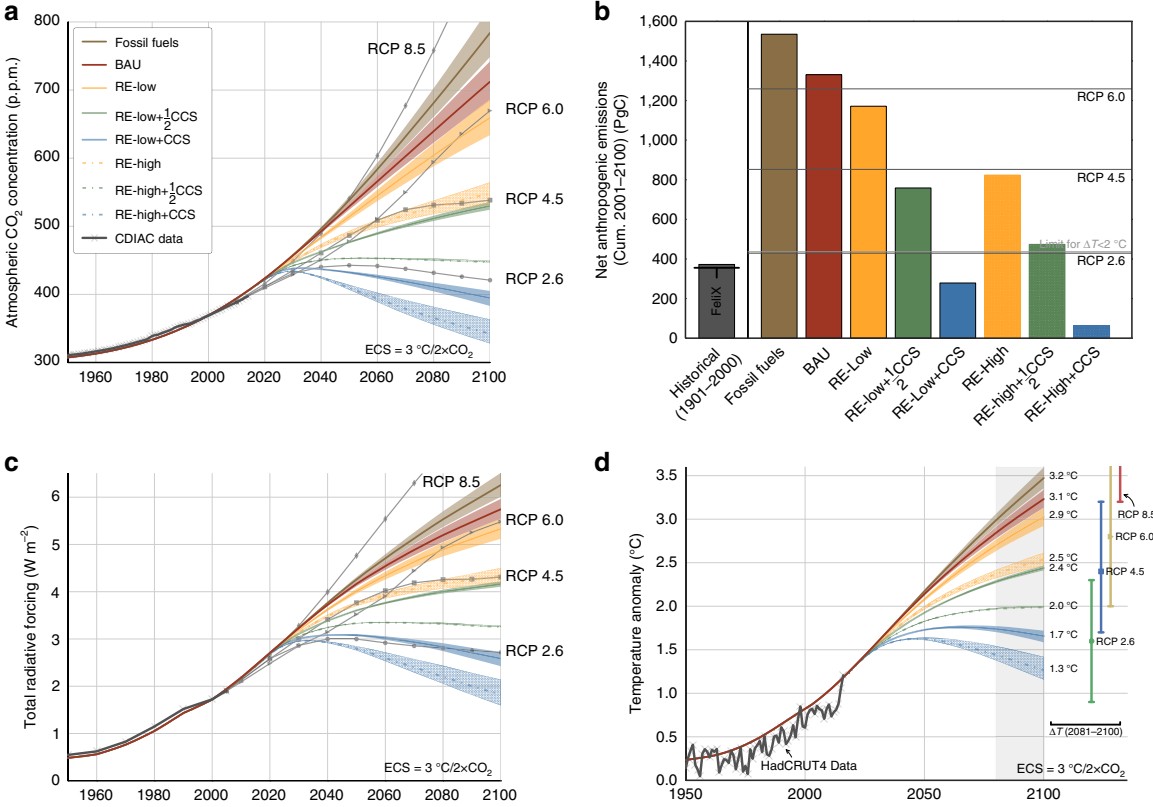

**Figure 6 | Global carbon-climate indicators.** (**a**) Atmospheric carbon concentration (p.p.m.), shown with CDIAC data[33] and RCP projections[7]; (**b**) cumulative anthropogenic emissions (2001–2100) [PgC], compared with historical emissions (IPCC[3] and FeliX) and RCP projections. (**c**) Total radiative forcing (W m$^{-2}$) for all greenhouse gases, shown with CDIAC data and RCP projections. $CO_2$ forcing is modelled endogenously; all other greenhouse gases are exogenously set to RCP 4.5. (**d**) Global average surface temperature change relative to preindustrial ($\Delta T$) (°C). Historical time series from HadCRUT4 (ref. 38). $\Delta T$ projections associated with each of the RCPs in 2100 are shown at right with 90% confidence intervals[40].

(that is, deliberately created as a result of mitigation policy) or natural (that is, caused inadvertently by the human perturbation of the carbon cycle). We have included natural sinks in this analysis to present a complete accounting of global carbon flows and because the carbon flux from the atmosphere into other parts of the Earth system (that is, biomass, soil and the oceans) will be essential to the magnitude and timing of economic and technological interventions necessary to realize the COP targets.

Second, carbon emissions pathways have an impact on natural sinks through carbon and temperature couplings in the FeliX model, but the sinks do not feed back into anthropogenic emissions from either the energy or LULUCF sectors. Although some mechanisms could complete this feedback loop (for example, permafrost thaw or carbon credits in a hypothetical global cap and trade market), they are not included in this analysis. For this reason, $R_{AF}$ separates the external, prescribed drivers of each scenario (that is, anthropogenic emissions) from the internal, dynamic responses (that is, natural sinks).

With its global scope, this analysis provides a physical science foundation for consideration of the more technically challenging and politically sensitive adaptation and finance goals of the Paris Agreement[1]. Similar to the four RCPs established as benchmarks by the IPCC, the scenarios included in this analysis cover a broad range of emissions pathways and can be used to generate insights into the magnitude and timing of the transitions needed to realize the COP emissions and warming targets[6]. In particular, our results show that the achievement of the COP emissions target ($R_{AF} \leq 0.0$) before 2100 will also be sufficient to limit warming below 2 °C, even after accounting

for the diminishing magnitudes of ocean and land sinks along decarbonizing pathways. This confirms that the COP emissions target should be regarded as the concrete benchmark for Intended Nationally Determined Contributions and subsequent negotiations.

Our analysis extends beyond the IPCC benchmarks in that the RCPs are not fully integrated scenarios of socio-economic, energy and LULUCF emission, and climate projections, but rather internally consistent pathways of atmospheric greenhouse gas concentrations. Indeed, the RCPs are intended as inputs to coupled emissions–carbon cycle–climate models such as FeliX[6]. Thus, although the scenarios in this analysis are not exogenously linked with specific RCPs, our reasons for drawing attention to points of comparison between the two sets are twofold: first, comparisons across multiple indicators (that is, emissions, radiative forcing and warming) serve to establish the internal consistency of the FeliX model. Second, comparisons between FeliX scenarios and the RCPs serve to link detailed, particular energy and LULUCF transitions with 'representative' pathways and mark progress towards the COP target.

The timing of peak energy sector emissions is a critical parameter for climate mitigation strategies. Even before the superposition of CCS, the RE-High scenario achieves much greater progress towards the 2 °C threshold than does the RE-Low scenario, largely because the former pathway accelerates peak emissions by over 30 years relative to the latter (*cf.* Table 1). Deeper and more rapid transformations of global agricultural and LULUCF systems can both cut emissions and increase net carbon flux into the land sink, further accelerating peak emissions.

Indeed, all of the scenarios examined here project progress towards curbing emissions from land use and land-use change, but none fully exploit the emissions mitigation potential of global agricultural and LULUCF systems[2,12,13].

As a set, our scenarios support broad characterizations of successful decarbonization strategies. Roughly speaking, and based on current technologies, energy sector emissions will need to peak within the next decade. By 2100, the market share of fossil fuels will need to fall to less than a fourth of total primary energy demand to preserve the possibility of meeting the COP targets.

The RE-High pathway (that is, sustained expansion of REs at a minimum of 5% per year) seems at least possible in light of recent trends. Since 1990, low-carbon technologies have grown at an annual rate of 2.2%, slightly outstripping annual growth in the total primary energy supply (1.9% per year)[14]. Over the same period, solar and wind energy have grown at 46.6% and 24.8% per year, respectively. However, this standout growth can be attributed to very low initial production rates. Although wind and solar infrastructures have increased by two to three orders of magnitude since 1990, their combined capacity in 2015 equalled roughly 1% of primary energy supply[14].

We also note that, when it is achieved in the $\frac{1}{2}$CCS and CCS emissions pathways and in similar analyses, full decarbonization relies on the coupling of CCS technology with bioenergy production, a carbon-negative process[15]. If coupling of CCS technology with bioenergy production is ultimately found to be unfeasible, uneconomical or unacceptably burdensome on ecosystems, then alternative negative emissions technologies (for example, direct air capture) will need to be substituted[16–18]. In the absence of these fail-safes, fossil fuels will need to be phased out completely and well before 2100.

Conversely, if the decarbonization of the energy and LULUCF sectors does not proceed apace, or if nutrient supply limits land sink carbon uptake[19,20], then carbon sequestration technology will need to be employed as quickly and as broadly as possible to meet the COP targets. Depending on the economic feasibility of carbon feedstocks for chemical production and chemical energy storage, permanent sequestration of up to an exagram of carbon (1,000 PgC) may be required. This may be technically possible, as global permanent storage capacity has been estimated to range from 135 PgC to as high as 2,700 PgC[21]. CCS on such a large scale remains a distant reality, given a current global annual geosequestration rate of only 7.3 MtC[10]. On the other hand, this low baseline makes sustained 30–35% year-over-year growth in the geosequestration rate a technologically realistic target, even before economies of scale begin to lower the energy and infrastructure costs of CCS.

Given the scientific community's rapidly evolving understanding of interlinkages among natural and economic systems[13], all mitigation options should be evaluated and it is appropriate to consider the impact of natural sinks on atmospheric carbon and warming pathways. At the same time, the flow of carbon into oceans, plants and soil will affect the functioning of these ecosystems and the services they provide. For example, persistent anthropogenic carbon emissions may affect agricultural yields and thermohaline circulation, as well as the life cycles and long-term stability of forest and marine ecosystems. The consequences of these eventualities, which are potentially as severe as those of global warming, can only be avoided by achieving the goal of strictly carbon-neutral societies as soon as possible.

## Methods

**Model scope and calibration.** FeliX models the complex interconnections among global human and natural systems to identify the probable economic and environmental impacts of trends, policies and technologies in the Anthropocene Era[22,23]. Fundamental linkages and feedbacks among demographic, economic,

land, energy, carbon and climate systems are drawn from published models, articles and sector reports, and codified as differential equations. These equations describe the status and flow of resources, subject to geoclimactic and economic parameters to characterize the present state and co-dependent development of natural and economic systems.

The model is calibrated to available historical data between 1900 and 2015 (refs 24,25). All FeliX model historical data, parameters, and results are calculated and reported as global averages.

**Non-CO$_2$ greenhouse gases.** Although the Paris Agreement encompasses all greenhouse gases, non-CO$_2$ emissions pathways are not modelled endogenously by FeliX. As a result, this analysis is limited to pathways for CO$_2$ mitigation. Emissions pathways and associated radiative forcing for non-CO$_2$ greenhouse gases (that is, CH$_4$, N$_2$O, HFC and 'others') are assumed to follow RCP 4.5, one of the IPCC's four benchmark pathways for atmospheric greenhouse gas concentrations, through 2100 (refs 6,7). The warming effects of alternative pathways for non-CO$_2$ emissions are plotted in Supplementary Fig. 4b.

**Population and GDP.** The model begins with the medium population projection from the United Nations Department of Economic and Social Affairs[26] and historical data on global gross domestic product (GDP) as an indicator of global economic growth[27].

**Energy sector.** Nominal energy demand per capita is linked sigmoidally with GDP per capita[28].

The RE-Low and RE-High scenarios are loosely calibrated to plausible energy futures under climate action scenarios, as identified by the Global Energy Assessment (MESSAGE model: geala_450_atr_nonuc, geaha_450_atr_full; IMAGE model: GEA_low_450 and GEA_high_450)[8]. In this way, we establish soft links between FeliX, and MESSAGE and IMAGE, two specialized models of global energy systems[8] (cf. Table 2).

Energy consumption in these pathways is compared with FeliX in Supplementary Table 3. In addition, we note that bioenergy production rates in the BAU and RE-Low scenarios are compatible with meta-analyses, including 'medium agreement' projections from the IPCC WG3, AR5 for year 2050, whereas bioenergy production in the RE-High scenario falls just beyond this range[29,30].

Although there are significant uncertainties associated with projecting energy consumption several decades into the future, primary energy profiles are not FeliX model results for the purposes of this analysis. Rather, they are definitional of their respective scenarios, and are intended to illustrate correspondences between $R_{AF}$ and energy sector transformations of various magnitudes.

We model low and high shifts in total primary energy demand as variations on the Fossil Fuels, BAU, RE-Low and RE-High scenarios. Primary energy consumption in 2100 in these scenarios is indicated by the columns to the right of PE profiles in Fig. 2. Carbon dioxide emissions from oil, gas, coal, biomass, solar, and wind energy are calculated as the product of total consumption and carbon intensities which reflect the full life-cycle of each technology[8]. $R_{AF}$ and warming projections are calculated for each of these variations and presented in Fig. 1 as shaded ranges around each of the central values.

In scenarios with CCS, the technology is implemented not as a step function, but as a sigmoidal expansion over the course of the twenty-first century to the maximum value of 40% ($\frac{1}{2}$ CCS) or 80% (CCS) of gross energy sector emissions. The expansion of CCS technology through 2100 is shown in Supplementary Fig. 1a.

**Kaya factors.** To facilitate comparison to other analyses, all FeliX scenarios are decomposed in Supplementary Fig. 6 into the four Kaya factors. In the population plot at top left, dotted lines indicate UNDESA high and low population variants[26]. In the GDP per capita (top right) and energy intensity of GDP (bottom left) plots, dotted lines project the continuation of trends from the recent past[31].

**LULUCF sector representation.** Land in the FeliX model is distributed among four mutually exclusive and collectively exhaustive categories: agricultural, forest, urban/industrial and 'other'. Each category is calibrated to FAOSTAT data on a global level (available for 1961–2010 for agricultural and 1990–2012 for forest and other land)[24]. Although not on a geographically explicit basis, land can be repurposed—most notably, due to changes in demand for agricultural land.

Land categorized as 'agricultural' is subdivided into arable land, permanent crops and permanent meadows and pastures. Arable land and permanent crops can be used to produce food, feed or energy crops, while permanent meadows and pastures are used only for feed production. The BAU scenario is calibrated to historical data available on FAO[24]. Crop and livestock yields are modelled endogenously as a function of input-neutral technological advancement, land management practices (that is, the expansion of high-input agriculture), water availability, pollution (including atmospheric carbon fertilization) and climate change.

Supplementary Fig. 1b plots FeliX model projections for crop yields in the BAU scenario. The model predicts an end to the steady expansion of agricultural land seen in the second half of the last century: through 2050, growth in demand for vegetal and animal products is likely to be satisfied by agricultural

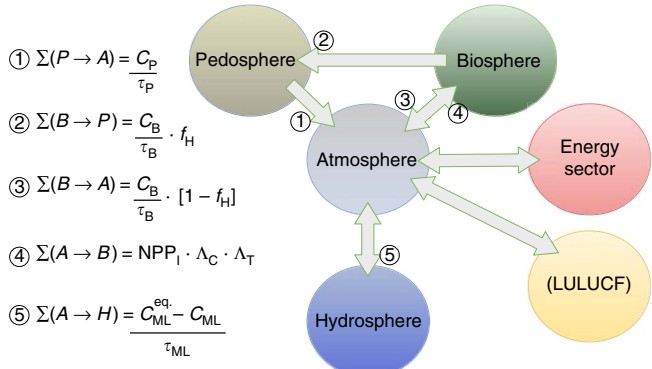

① $\Sigma(P \rightarrow A) = \dfrac{C_P}{\tau_P}$

② $\Sigma(B \rightarrow P) = \dfrac{C_B}{\tau_B} \cdot f_H$

③ $\Sigma(B \rightarrow A) = \dfrac{C_B}{\tau_B} \cdot [1 - f_H]$

④ $\Sigma(A \rightarrow B) = NPP_I \cdot \Lambda_C \cdot \Lambda_T$

⑤ $\Sigma(A \rightarrow H) = \dfrac{C_{ML}^{eq.} - C_{ML}}{\tau_{ML}}$

**Figure 7 | Schematic illustration of gross flows through the global carbon cycle.** At left: FeliX model formulas for calculating gross carbon flux from reservoir X to reservoir Y, $\Sigma(X \rightarrow Y)$ (PgC per year). FeliX model parameters are based on the C-ROADS model[34] and are defined and discussed in further detail in the model's technical documentation[28].

intensification. After midcentury, however, the cumulative effects of fertilizer saturation, water scarcity and ozone pollution may cause agricultural yields to stagnate or decline. As demand for food (in particular animal products) continues to grow, agricultural land may begin to expand indefinitely after 2050 at the expense of natural habitats.

Between 2010 and 2100, both supply- and demand-side trends lead to a 17% expansion in agricultural land. As land is a finite resource, transitions are zero sum. In the BAU scenario, agricultural land expands at the cost of natural habitats included in forests and 'other' land (that is, grassland). The general category of 'forest' includes both natural and managed stands, although emissions from conversion from natural to managed forest (and vice versa) are explicitly accounted for ref. 32.

**LULUCF sector emissions.** $CO_2$ emissions from the LULUCF sector include deforestation and forest conversion to managed forests and plantations, net of afforestation. All processes are calculated explicitly in the model as the net effect of global population growth, per capita energy and food demand, and agricultural yields[2,32]. Relative to energy sector emissions, LULUCF emissions are subject to very large uncertainties (cf. Fig. 4), as shown by divergences among annual and cumulative LULUCF emissions estimates from the IPCC, CMIP5, CDIAC and RCP database.

In the FeliX model, LULUCF emissions per unit area are calibrated using deforestation rates from the FAO[24] and historical $CO_2$ emissions from CDIAC[33]. This method produces a historical emissions pathway well within the error range as estimated by IPCC and CMIP5 analyses[3,5]. Future LULUCF emissions projections fall within the full range of RCP projections.

In the Fossil Fuels scenario, LULUCF emissions drop to zero at the end of the century as easily-deforested areas are depleted and the persistent predominance of fossil fuels limits land-use change for bioenergy production. Greater bioenergy demand in the BAU scenario leads to greater land-use change and associated emissions.

Much of the land-use change in the Fossil Fuels and BAU scenarios can be attributed to 'unnecessary' deforestation, which is forest loss from failure to optimize land use in ways that are technically possible. Unnecessary deforestation, an example of squandered resources, is land-use change driven by social and political constraints, including conflict, poor governance, perverse incentives, poverty and shortage of labour or capital[9]. Consistent with all four RCPs, the RE-Low and RE-High scenarios eliminate unnecessary deforestation based on the assumption that any significant future transition to REs will be coupled with enhanced protection of terrestrial landscapes and their carbon stocks[3].

The nominal rate of input-neutral growth is shifted up and down to evaluate the impacts of alternative yields. The magnitude of these shifts is indicated by the shaded region in Supplementary Fig. 1b, but the effect of this shift is smaller than that of primary energy demand, and is therefore suppressed throughout figures in the main paper.

**Carbon reservoir flux.** The model calculates projected $CO_2$ emissions based on representations of carbon emissions from the energy and land-use change sectors, as discussed above. These emissions accumulate in the atmosphere until they are absorbed into the biosphere, pedosphere or oceans based on C-ROADS, a Simple Climate Model, which has been used extensively for climate policy impact analysis and decision making by parties to the UNFCCC[34,35]. Pathways and simplified equations for gross carbon flux among the reservoirs are illustrated in Fig. 7 and discussed below[28].

Supplementary Figs 2 and 3 illustrates the ocean and land sink responses to increasing atmospheric carbon concentration. In Supplementary Fig. 2, cumulative uptake is plotted for both natural sinks relative to atmospheric carbon. In Supplementary Fig. 3, we plot the absolute (a) and fractional (b) responses of net sink flux to a range of constant emissions rates. For example, Supplementary Fig. 3b indicates that a 100% increase in annual emissions generates a 150% increase in net atmospheric carbon flux, an 80% increase in net carbon uptake by the land sink, and only a 50% increase in oceanic carbon uptake. Conversely, a 50% decrease in anthropogenic emissions generates a 66% decrease in net atmospheric carbon flux and 50 and 35% decreases in net land sink and ocean fluxes, respectively.

For the nominal ECS value ($3.0\,°C/2 \times CO_2$), FeliX projects a transient climate sensitivity of $2.4\,°C/2 \times CO_2$. For $ECS = 2.5\,°C/2 \times CO_2$, the transient climate sensitivity is $2.1\,°C/2 \times CO_2$.

**Land sink carbon flux and feedbacks.** Carbon flow from the atmosphere into the land sink begins from preindustrial net primary productivity (initial NPP = 85.2 PgC per year). This initial carbon flux is coupled with atmospheric carbon concentration and global surface temperature via bio-stimulation ($\Lambda_C$) and climate ($\Lambda_T$) coefficients, respectively.

The bio-stimulation feedback mechanism introduces logarithmic NPP-carbon coupling as described in equation (1) ($\lambda_C = 0.35$). This mechanism increases land sink carbon uptake by 1.25 GtC p.p.m.$^{-1}$ (cf. Table 3) such that a doubling of the carbon content of the atmosphere ($C_A$) relative to preindustrial ($C_A^{PI}$) generates a 24% increase in NPP before taking into account other feedbacks[20].

$$\Lambda_C = 1 + \lambda_C \cdot \ln\left(\frac{C_A}{C_A^{PI}}\right) \qquad (1)$$

Within the land sink, the carbon stocks of the biosphere (plants) and pedosphere (soil) are modelled separately to account for distinct characteristic residence times. Gross carbon flow out of biomass is equal to biomass carbon stock ($C_B$) divided by a constant residence time ($\tau_B = 10.6$ years). Treating this parameter as a constant neglects the impact of water and other nutrient availability in unmanaged ecosystems, which are not represented in the FeliX model[20]. This flow is distributed between the pedosphere and the atmosphere according to the constant biomass humification fraction ($f_H = 0.428$; cf. Fig. 7 (2 and 3)). Gross carbon flow out of the pedosphere is equal to the carbon content of the reservoir ($C_P$) divided by its residence time ($\tau_P$). This residence time is coupled with climate, as discussed below.

Carbon flux through the land sink is linked with global temperature change at two points in the model, providing feedback to both NPP and pedosphere residence time. NPP–climate coupling is included in the nominal land sink carbon flux through the climate coefficient ($\Lambda_T$ in Fig. 7 (4) and equation (2) below), where the coefficient ($\lambda_T = 0.012\,K^{-1}$) is calibrated to match the average value in literature of the land sink-climate feedback mechanism[4,5].

$$\Lambda_T = 1 - \lambda_T \cdot \Delta T \qquad (2)$$

Second, the average residence time of carbon in soil ($\tau_P^{PI} = 27.8$ per year for preindustrial climate; cf. Fig. 7 (1)) is linearly coupled with climate, as shown in equation (3) below, where $\lambda_P = 0.5$ year K$^{-1}$.

$$\tau_P = \tau_P^{PI} - \lambda_P \cdot \Delta T \qquad (3)$$

The net effect of the land sink-climate coupling reduces land sink carbon uptake by 66 PgC K$^{-1}$ (cf. Table 3)[4]. In year 2050 of the BAU scenario, this effect reduces global soil carbon reserves by 26 PgC (measured relative to BAU without the coupling). This figure is in line with the latest global estimates ($30 \pm 30$ PgC)[36].

Net land sink carbon flux is plotted for all scenarios in Fig. 5c. The cycling and availability of nutrients including N, P and water represent another important feedback to land sink carbon flow in general and NPP in particular[20]. These considerations tend to limit NPP response to atmospheric carbon concentrations and should be included in subsequent iterations of this analysis.

In deep decarbonization scenarios, net carbon flux into the ocean and lands reservoirs switches directions, turning these sinks into carbon sources. The simplest physical explanation for this effect is that rapid decarbonization reverses the sign, or direction, of the chemical coupling between the atmosphere, ocean and land sink. This coupling is responsible for increasing the net carbon flux into natural sinks as emissions rise; thus, it should also be expected to cause a net carbon flow out of these sinks if net-negative anthropogenic emissions are achieved and atmospheric carbon concentration begins to drop.

From another perspective, we could invoke Le Châtelier's principle to predict that the earth system will act to 'resist' change. In deep decarbonization scenarios, the 'change' is the reduction in atmospheric carbon concentration resulting from net-negative anthropogenic emissions and the natural 'response' is the net flow of carbon out of the land and ocean sinks.

**Ocean carbon flux and feedbacks.** In addition to cycling through terrestrial reservoirs, carbon is removed from the atmosphere through dissolution into the mixed ocean layer (depth 0–100 m) and subsequently propagates through four independently modelled deeper layers (100–400, 400–700, 700–2,000 and 2,000–2,800 m).

**Table 1 | Expansion of REs.**

|  | Annual growth (%)* | | | Net em. peak in | $\Delta T$ (°C)† (2100) |
|---|---|---|---|---|---|
|  | **Wind** | **Solar** | **Biomass** |  |  |
| Fossil Fuels | 1.8 | 3.6 | 1.8 | 2099 | 3.5 |
| BAU | 3.1 | 5.3 | 3.7 | 2054 | 3.2 |
| RE-Low | 3.9 | 6.1 | 3.7 | 2048 | 3.1 |
| RE-High | 4.7 | 6.8 | 3.9 | 2022 | 2.5 |

BAU, business-as-usual; ECS, equilibrium climate sensitivity; RE, renewable energy.
Expansion of REs assuming constant geometric growth through 2100, starting from 2013 IEA base values[14]. The fourth column indicates the year in which total net anthropogenic emissions peak in each scenario, and the final column lists $\Delta T$ projections for each scenario for ECS = 3.0 °C/2 × CO_2.
*2013 basis: wind: 2.30 EJ per year; solar: 1.68 EJ per year; biomass (utility scale): 8.33 EJ per year.
†ECS = 3.0 °C × 2 × CO_2.

**Table 2 | Primary energy consumption in year 2100 of FeliX and similar models.**

Primary energy (EJ per year) (2100)

| Model (scenario) | Coal | Oil | Gas | Biomass | Solar | Wind | Nuc. & hydro. | Total |
|---|---|---|---|---|---|---|---|---|
| MESSAGE | 41-74 | 2-3 | 46-65 | 221 | 250-327 | 34-89 | 23-284 | 614-1051 |
| IMAGE | 93-360 | 63-75 | 178-181 | 216-220 | 28-31 | 16-47 | 26-201 | 630-1106 |
| FeliX (Fossil Fuels) | 155-209 | 219-249 | 473-640 | 29-58 | 26-50 | 8-15 | 58 | 968-1279 |
| FeliX (BAU) | 110-143 | 150-160 | 323-379 | 155-268 | 114-196 | 24-41 | 58 | 934-1245 |
| FeliX (RE-Low) | 74-99 | 116-162 | 266-323 | 168-231 | 223-365 | 50-81 | 58 | 956-1319 |
| FeliX (RE-High) | 20-29 | 20-70 | 112-147 | 215-276 | 425-603 | 104-145 | 58 | 954-1328 |

Primary energy consumption in year 2100 of FeliX and similar models. For FeliX, ranges are defined by low and high shifts to nominal primary energy demand (cf. Fig. 2). MESSAGE ranges include geala_450_atr_nonuc and geaha_450_atr_full scenarios, and IMAGE ranges include the GEA_low_450 and GEA_high_450 scenarios[8].

**Table 3 | Parameters describing chemical and climate feedback to land and ocean sinks.**

|  | $\alpha$ | $\beta_L$ | $\gamma_L$ | $\beta_O$ | $\gamma_O$ | Gain |
|---|---|---|---|---|---|---|
|  | [(K)/(p.p.m.)] | [(GtC)/(p.p.m.)] | [(GtC)/(K)] | [(GtC)/(p.p.m.)] | [(GtC)/(K)] | — |
| FeliX | 0.0067 | 1.25 | − 66 | 1.23 | − 46 | 0.21 |
| C4MIP ensemble average value | 0.0061 | 1.35 | − 79 | 1.13 | − 30 | 0.15 |
| C4MIP ensemble low value | 0.0038 | 0.20 | − 177 | 0.80 | − 67 | 0.04 |
| C4MIP ensemble high value | 0.0082 | 2.80 | − 20 | 1.60 | − 14 | 0.31 |

C4MIP, Coupled Climate Carbon Cycle Model Intercomparison Project.
Parameters describing chemical and climate feedback to land and ocean sinks climate sensitivity ($\alpha$), land sink sensitivity to carbon ($\beta_L$) and climate ($\gamma_L$) and ocean sensitivity to carbon ($\beta_O$) and climate ($\gamma_O$). All parameters are calculated as in C4MIP[4]. This meta-analysis is also the source of the averages in the second row, which report directly comparable feedback parameters for an ensemble of 11 similar models.

The equilibrium dissolved inorganic carbon content of the mixed ocean layer ($C_{ML}^{eq}$) is given by equation (4), where $C_{ML}^{PI}$ is the preindustrial carbon content of the oceans, $C_A$ is the present carbon content of the atmosphere and $C_A^{PI}$ is the preindustrial carbon content of the atmosphere. The carbon content of the mixed layer ($C_{ML}$) is assumed to reach equilibrium with the atmosphere with a constant characteristic mixing time of 1 year ($\tau_{ML} = 1$ year).

The ocean-climate coupling ($\Lambda_O$; $\lambda_O = 0.0045$ K$^{-1}$; cf. equation (5)) reduces carbon uptake by 46 GtC K$^{-1}$ (cf. Table 3).

Finally, the ocean-carbon coupling is expressed by the dimensionless Revelle factor ($\zeta$), which expresses the marginal capacity of the oceans to absorb carbon ($\delta_R = 0.0045$). The Revelle factor increases logarithmically with the carbon content of the oceans, rising from its initial value ($\zeta_I = 9.7$) to 10.9 in year 2010 of the simulation (cf. equation (6))[4]. The ocean-carbon coupling increases carbon uptake by 1.23 GtC p.p.m.$^{-1}$ (cf. Table 3).

$$C_{ML}^{eq} = C_{ML}^{PI} \cdot \Lambda_O \cdot \left( \frac{C_A}{C_A^{PI}} \right)^{1/\zeta} \quad (4)$$

$$\Lambda_O = 1 - \lambda_O \cdot \Delta T \quad (5)$$

$$\zeta = \zeta_I + \delta_R \times \ln\left( \frac{C_A}{C_A^{PI}} \right) \quad (6)$$

**Climate gain.** Table 3 presents FeliX model parameters including climate sensitivity ($\alpha$), land sink sensitivity to carbon ($\beta_L$) and climate ($\gamma_L$), and ocean sensitivity to carbon ($\beta_O$) and climate ($\gamma_O$), calculated for 2100 as in the Coupled Climate Carbon Cycle Model Intercomparison Project (C4MIP)[4,5]. Overall gain (g) of the climate system, which quantifies the ratio of temperature change due to these feedback loops to total temperature change, is shown in the column at the far right[4,37]. For comparison to similar models, we also show the average value for each parameter from the ensemble of 11 models included in the same study. All FeliX parameters show satisfactory agreement with the carbon flux drivers and feedbacks as modelled in this ensemble and in the subsequent iteration, CMIP5 (ref. 5).

**Temperature anomalies.** HadCRUT4 data on global surface temperature anomaly are used for results validation, and represent observed temperature increases relative to the period (1850–1900) from the Met Office Hadley Center[38].

Global surface temperature change is affected by radiative forcings, feedback cooling due to outbound longwave radiation, and heat transfer from the atmosphere and mixed ocean layer to the four deep ocean layers. Net radiative forcing is calculated from the concentration of carbon in atmosphere, a product of $CO_2$ emissions from the energy and LULUCF sectors and other greenhouse gases, including $CH_4$, $N_2O$, halocarbons, and other gases and aerosols. Endogenous projections of atmospheric carbon concentration are used to model associated radiative forcing anomaly for carbon dioxide. Forcing anomalies associated with

other greenhouse gases are modeled exogenously using RCP 4.5 (ref. 7). Total greenhouse gas forcing is translated into temperature anomalies as in the C-ROADS model[34,35] (*cf.* Fig. 6).

A negative feedback loop incorporates heat transfer from the atmosphere and the upper ocean into space via outbound longwave radiation. The magnitude of this feedback, or cooling, is determined by the ECS, a metric used to characterize the response of the global climate system to a given forcing. ECS is broadly defined as the equilibrium global mean surface temperature change following a doubling of atmospheric $CO_2$ concentration. In the FeliX model, ECS is nominally set equal to $3.0\,°C/2 \times CO_2$. In Supplementary Fig. 4a, temperature anomaly projections are shown for the BAU scenario over the full range of probable values for ECS (1.5–4.5) as identified by the IPCC[11]. In Supplementary Fig. 5, we plot $R_{AF}$ and $\Delta T$ projections based on $ECS' = 2.5\,°C/2 \times CO_2$. This range of values is consistent with Coupled Climate Carbon Cycle Model Intercomparison Project results[4] and the latest estimates from IPCC[11].

**$R_{AF}$ definition and error propagation.** We begin by identifying carbon sources X = {FF, LUC, RE} and sinks Y = {O, LS} as shown in Supplementary Table 1. Net annual carbon flux from each emissions source (X) is labeled $\Upsilon_X$, while net annual flux into each reservoir (Y) is labeled $\Omega_Y$. For all sources (X), positive values of $\Upsilon_X$ indicate net positive emissions. For all reservoirs (Y), positive values of $\Omega_Y$ indicate net uptake of carbon. Net annual increases in atmospheric carbon are defined as the sum of emissions and sink fluxes as shown in equation (7):

$$\Omega_{Atm} = \Upsilon_{FF} + \Upsilon_{LUC} + \Upsilon_{RE} - \Omega_O - \Omega_{LS} \quad (7)$$

In accordance with the COP21 text, we define $R_{AF}$ in equation (8):

$$R_{AF} = \frac{\Upsilon_{FF} + \Upsilon_{LUC} + \Upsilon_{RE}}{\Omega_O + \Omega_{LS}} \quad (8)$$

This equation is used to calculate $R_{AF}$ for all scenarios, as plotted in Fig. 1. We calculate the error on $R_{AF}$ in the standard manner:

$$\Delta R_{AF}^2 = \frac{\partial R_{AF}^2}{\partial \Upsilon_{FF}} \Delta \Upsilon_{FF}^2 + \frac{\partial R_{AF}^2}{\partial \Upsilon_{LUC}} \Delta \Upsilon_{LUC}^2 + \frac{\partial R_{AF}^2}{\partial \Upsilon_{RE}} \Delta \Upsilon_{RE}^2 \\ + \frac{\partial R_{AF}^2}{\partial \Omega_O} \Delta \Omega_O^2 + \frac{\partial R_{AF}^2}{\partial \Omega_{LS}} \Delta \Omega_{LS}^2 \quad (9)$$

Taking the partial derivatives:

$$\frac{\partial R_{AF}}{\partial \Upsilon_{FF}} = \frac{\partial R_{AF}}{\partial \Upsilon_{LUC}} = \frac{\partial R_{AF}}{\partial \Upsilon_{RE}} = \frac{1}{\Omega_O + \Omega_{LS}} \quad (10)$$

$$\frac{\partial R_{AF}}{\partial \Omega_O} = \frac{\partial R_{AF}}{\partial \Omega_{LS}} = \frac{\Upsilon_{FF} + \Upsilon_{LUC} + \Upsilon_{RE}}{[\Omega_O + \Omega_{LS}]^2} \quad (11)$$

We arrive at:

$$\Delta R_{AF}^2 = \frac{1}{[\Omega_O + \Omega_{LS}]^4} \left\{ (\Omega_O + \Omega_{LS})^2 \cdot \left[ \Delta \Upsilon_{FF}^2 + \Delta \Upsilon_{LUC}^2 + \Delta \Upsilon_{RE}^2 \right] \\ + (\Upsilon_{FF} + \Upsilon_{LUC} + \Upsilon_{RE})^2 \cdot \left[ \Delta \Omega_O^2 + \Delta \Omega_{LS}^2 \right] \right\} \quad (12)$$

We use equation (12) to calculate errors on $R_{AF}$ using the relative errors measured by the IPCC (*cf.* Supplementary Table 1). This is a very conservative projection, given the probable advancement of global carbon monitoring technologies and techniques. For scenarios with CCS, errors are calculated on gross emissions from REs.

**Data availability.** The most recently published version of the FeliX model is freely available for download and use at the model website[32]. The version of the model used for this analysis will be made available on the same site upon publication of the manuscript. The authors agree to make all scenarios used in this analysis available upon request.

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

## Acknowledgements

We acknowledge support from the European Research Council Synergy grant ERC-2013-SyG-610028 IMBALANCE-P and the foundational work on the FeliX model done by Dr Steffen Fritz, Dr Ian McCallum and Florian Kraxner of IIASA. We also thank Dr Joeri Rogelj for reviewing and offering helpful comments on the manuscript.

## Author contributions

B.W. and M.O. conceived of the analysis. F.R. originated the FeliX model with M.O. and B.W. continued its development. B.W. performed the analysis and was responsible for drafting the manuscript and generating all figures. P.C., I.J., J.P., K.R., F.R. and D.v.V. contributed substantially and equally to the conception of the analysis, the original manuscript and its subsequent revision. M.O. oversaw the paper as group leader and senior author.

## Additional information

**Competing interests:** The authors declare no competing financial interests.

**Publisher's note**: 

