## [Peer Review File · Nature Communications]

Reviewers' comments:

Reviewer #1 (Remarks to the Author):

This paper explores a number of scenarios of energy use and land management to see the kinds of initiatives needed to meet the objectives of the Paris agreements from COP 21. They define an "atmospheric flux ratio" and use it to show that net negative emissions are really required to meet the COP objectives for this century. It is a broadly interesting and important topic.

The paper tries to cover a huge amount of information and analysis in a very short text and as a consequence it becomes very terse and is demanding for the reader. I found the writing to be ok, but not as smooth as might be desired.

I find it difficult to express a comprehensive evaluation of the paper so let me move through it sequentially and comment on some of the specifics or examples of what challenged me.

Paragraph 2 defines an "atmospheric flux ratio" and then talks about "emissions ratios". I think these are the same thing and consistent vocabulary would help. It then adds that the atmospheric flux ratio is the inverse of the airborne fraction, which is not consistent with my definitions of inverse and airborne fraction.

Paragraph four has a couple of hyphens which are problematic. This punctuation occurs several times in the text. It appears that commas or parentheses would be more clear.

In the last sentence of paragraph 5 I think we infer that there is a complete linearity between temperature and the CO₂ increase and that a doubling of preindustrial CO₂ would lead to a temperature increase of 3 degrees C.

In the results section, line 1, presumably the IPCC data are found in reference 2. In general the first 2 paragraphs of this section are verbally complex and awkwardly written.

Page 3, first full paragraph, I think we do not have a shortage of poverty. I do not mean to pick specifically on this sentence but I think it is symptomatic of some of the awkward writing throughout the paper. See also the word "standalone" in the figure caption above.

Figure 1 well illustrates the challenge of presenting a great deal of information in a single picture. The legend is difficult to read but apparently the hydro and nuclear band is in yellow, but it is not clear what the black line of IEA data represents. Presumably the market share and atmospheric CO₂ data at the bottom of each figure are time linked (at 25 year increments?)

Figure 2 is not adequately described. The colors of so many lines are hard to distinguish, the "ECS" is defined only 1 place in the text, the "CCS25" is defined nowhere. The brown, red, and orange vertical bars seem clear, the blue and green do not. The points with apparent error bars are not described at all.

On page 4 the paragraph on "primary energy demand" is poorly connected to the rest of the text and poorly explained. Or maybe it is just in the wrong place in the text.

Figure 3 has challenges similar to those in figure 2. The little dots with error bars are not explained, it takes some digging to understand why there appear to be 2 lines for the historical data, the legend does not appear until Figure 3b, and one really needs some expertise in the field to understand the appearance of the RCP lines. My goodness there is a lot of information in Figure 3. This is not for the faint of heart.

On page 6 the first partial paragraph has text "this mitigation can up to a certain point be achieved..." which is very unclear. I thought that all of this was already rolled into the scenarios. So what are we saying here?

Page 7, the first sentence under the heading "net negative.." uses the word "more", more than what? The same paragraph gets to the idea shown in Figure 3c that "negative atmospheric carbon flux ultimately leads to a net outflow of carbon from plant, soil, and ocean reservoirs..." This becomes sufficiently important in the conclusions that I suggest it needs some explanation of the physics involved. My understanding is that net flows between the ocean and atmosphere depend on the concentration gradient and thus only indirectly on anthropogenic emissions.

In reference 7 I suspect that author Van Vuuren would like to have the Van with his name here.

I did not read all of the way through the supplemental material.

My overall reaction is that for such a complex paper to be accessible to a wide audience it needs a smoothly flowing text and meticulous attention to detail.

Reviewer #2 (Remarks to the Author):

This paper analyzes a series of alternative greenhouse gas emission and CO₂ removal pathways to limit global warming. It focusses on one of the goals agreed by the Conferences of Parties (COP) in the recent Climate Conference in Paris in December 2015: "... to achieve a balance between anthropogenic emissions by sources and removals by sinks of greenhouse gases in the second half of this century." The goal was introduced in support of the primary climate goal that for many years has been the central subject of the COP conferences: the limitation of global warming to maximally 2deg C. The relation between the recent Paris balanced-source-sink goal and the long-standing COP 2 deg C global warming goal is an interesting scientific question, with considerable practical political implications. It can be expected to appeal strongly to the general reader. Unfortunately, however, the inter-relationship is not discussed in detail in the paper, but referred to only marginally. There is also no reference to other attempts to translate the 2 deg C goal into practical political implications, such as the limitation this imposes on the maximal global per capita CO₂ emissions.

These shortcomings reflect a general tendency of the authors to follow their own path, without detailed reference to other work. Thus the new set of emission pathways analyzed by the authors, although of considerable interest in their own, are not tied in to the extensive literature that already exists on climate mitigation computations. There are indeed references (7,8) to the "representative concentration pathways" project, a new major international effort to coordinate mitigation analyses, and figures 3 and 4 actually include RCP data as "benchmarks". But there is no explanation this reviewer could discover, either in the figure or elsewhere, of how the RCP data relate to present paper (apart from a cryptic comment "... resulting in an emissions pathway that is comparable to RCP 4.5..." in the last paragraph before the Discussion section).

A minor comment, but in the same vein: it would be helpful for the general reader to have better explained why the authors prefer to introduce as the basic variable of their paper the atmospheric flux ratio $RAF = x/y$, where $x = \text{net anthropogenic CO}_2 \text{ emissions}$ and $y = \text{net CO}_2 \text{ uptake by natural sinks}$, rather than the familiar airborne fraction $AF = (x - y)/x$. RAF is clearly not "the airborne fraction inverse" as stated by the authors (second paragraph). The rest of the paragraph nonetheless clearly explains the useful information content of RAF. But the first two paragraphs of the following section "Results" are again less clear. The relation between x and y is better described, not in terms of "feedback", but by the fact that while x is an externally prescribed variable at any given instant, y represents an integral system response dependent on the past history of the system forcing x , and will thus always adjust in time to any change in x . There is no feedback of y on x .

Apart from these general comments - which can hopefully be responded to without major modifications - the paper represents a very useful and readable analysis of the impact of various feasible mitigation strategies in response to the secondary COP 21 goal of neutralizing carbon emissions by the end of this century. The figures are well organized and clear, and the impact of different combinations of the various levels of renewable-energy and carbon-capture-and-sequestration technologies well presented.

Of interest, in particular, is the critical role of carbon capture and sequestration, which needs to be

introduced on a major scale in addition to renewable energy to meet the COP 21 goals. This reviewer would have welcomed a more detailed discussion of the political implications of the role of CCS, which is still in its infancy regarding large scale implementation compared to that of renewable energy, which, despite its present low level of global implementation, is technically more advanced. However, it is understandable that the authors may prefer to leave a more detailed discussion of this important question to follow-up publications.

In summary, I recommend publication with minor modifications.

Klaus Hasselmann, em.Dir., Max Planck Institute for Meteorology, Hamburg

Reviewer #3 (Remarks to the Author):

General comments

This manuscript describes an analysis of the effects of a variety of future scenarios of emissions, land use change, and technology change, oriented in particular around the ratio of anthropogenic emissions to natural sinks. This is potentially interesting, especially given that the authors are using a relatively new model and non-standard (i.e. not RCP) scenarios. The ms is well written, easy to read, and generally interesting.

There are a number of issues that concern me here, however. First, I'm unconvinced by the utility of Raf, the authors' metric of choice; it seems confusing and difficult to interpret, or at least vulnerable to misinterpretation. See specific comment below.

Second, Felix is not well known in the integrated assessment world (NB I am not an integrated assessment modeler, but am reasonably familiar with the field), and a bit more information in the text about its design and limitations would be useful. For example, as far as I can tell its uses relationships, based on historical information, between key parameters that are then forecast forward. It doesn't seem to have any technological detail, process detail, geographic regions, nor any physical constraints (e.g., land area for food, oil resources). It would be good to discuss this a bit, acknowledging limitations and strengths—i.e. why you chose this model design.

Third, some of the physical parameters or characteristics of the model are not well documented and seem questionable (see specific comments below). In particular, Felix's assumption of preindustrial NPP is 50% higher than most other estimates, if I understand correctly, which raises doubts about the fidelity of the model's entire C cycle. This needs to be clarified/justified/explained much better.

Finally, the discussion spends too much time restating results, and doesn't really compare these results to previous, similar studies, or provide much in the way of context. As a result the novelty of this work is not clear.

In summary, there are many points of interest here, but the current ms needs extensive revisions to better explain and justify the model choice, parameters, Raf ratio, and better contextualize the results with respect to previous work in this field.

Specific comments

1. In the future please number lines—makes reviewing much easier

2. Page 1: why is defining an "Raf" ratio necessary, useful, or appropriate? After all, the effects of atmospheric CO₂ are determined by its absolute level in the atmosphere, not the ratio of inflows to outflows; and a ratio is also subject to distortion at, e.g., small flux levels (as it's nonlinear with respect to flux size), which makes interpreting it confusing. So I think that this needs better, clearer justification
3. P. 4: "and climate" not "an climate"
4. P. 9: preindustrial terrestrial NPP of 85 Pg C/yr? That's really high—most ESMs, remote-sensing analyses, etc., put current NPP much lower. For example, the CASA model simulates ~48 (<http://www.sciencedirect.com/science/article/pii/S003442579400066V>); Nemani et al. estimated ~54 (<http://science.sciencemag.org/content/300/5625/1560>); FLUXNET syntheses, etc., all produce numbers in the 46-56 range as far as I'm aware. Where did this 85 come from?
5. P. 9: And where did these turnover times (10.6 years for biomass, 27.8 years for soil) come from? Are they consistent with the most recent, best estimate I'm aware of, Carvalhais et al. 2014 (10.1038/nature13731)?
6. P. 10: "emerging results" from CMIP5? Isn't CMIP5 pretty well 'done' at this point?
7. P. 10: What exactly does "gain" in table 2 mean?

Reviewer #4 (Remarks to the Author):

- A. The authors propose a new assessment of historic and future global atmospheric emissions of carbon using the FeliX model. The model takes into account economic and natural parameters to compute the annual stocks and flows (emissions and sinks) of carbon.
- B. This study is original in the sense that it considers the impact of human activities and natural sinks in the global carbon cycle, and relies on the ratio of the first over the latter (the atmospheric flux ratio R_AF) to quantify pathways for various policy options. This indicator, as well as carbon emissions and temperature anomalies, are computed until 2100.
- C. Most data can be deemed robust as it comes from international bodies such as IEA, and as such there is a consensus on its validity. Similarly, the scenarios tested are consistent with each other.
- D. Uncertainty analysis is crucial in this type of exercise, especially for such long-term storylines. As far as I can see, scenarios and their variants account for enough uncertainty for the results to be considered robust. All figures are reported with confidence intervals, and sensitivity analyses and error analyses are performed for all scenarios. Clearly, including feedbacks in climate modelling introduces major uncertainty, but the authors have made serious efforts to understand and quantify it.
- E. The study is robust overall, and can be seen as a valuable addition to existing knowledge on the contribution of anthropogenic carbon emissions to the carbon cycle. The results are surprisingly optimistic, as the consensus tends to be that 2°C warming objectives are becoming unattainable, which makes the conclusions of this study very interesting. Of course, increasing decarbonization of the anthroposphere is necessary, but gives hope concerning the COP emissions target.
- F. The work looks very much final, and I do not have much to say about improvements. I would recommend the publication of this work without significant change to the manuscript. More scenarios could of course always be included, but keeping the sources limited to the IEA and IPCC provides an overall consistency and robustness - and I believe that FeliX is flexible enough to test other scenarios easily.
- G. The list of references may remain unmodified.

H. The manuscript is clear and provides factual information throughout the various sections, I could not find a claim that came unsubstantiated.

Reviewer #1 (Remarks to the Author):

This paper explores a number of scenarios of energy use and land management to see the kinds of initiatives needed to meet the objectives of the Paris agreements from COP 21. They define an “atmospheric flux ratio” and use it to show that net negative emissions are really required to meet the COP objectives for this century. It is a broadly interesting and important topic.

The paper tries to cover a huge amount of information and analysis in a very short text and as a consequence it becomes very terse and is demanding for the reader. I found the writing to be ok, but not as smooth as might be desired.

I find it difficult to express a comprehensive evaluation of the paper so let me move through it sequentially and comment on some of the specifics or examples of what challenged me.

Thank you for this frank and useful assessment. In addition to addressing the specific comments that follow, we have reworked and expanded the text throughout for clarity and precision. Changes from the original draft are highlighted in the revised manuscript. Where specific changes are referenced in the following discussion, we include line numbers for convenience.

The reviewer offered several useful suggestions for making the figures throughout the analysis more accessible to the reader, and much of the work on this revision has gone into refining the figures. Visual clutter was identified as a major issue, so we have switched from 3 CCS rates (25%, 50%, and 75% reduction to gross energy emissions) to 2 (40% and 80%). We submit that this makes the figures easier to interpret without loss of information.

Paragraph 2 defines an “atmospheric flux ratio” and then talks about “emissions ratios”. I think these are the same thing and consistent vocabulary would help. It then adds that the atmospheric flux ratio is the inverse of the airborne fraction, which is not consistent with my definitions of inverse and airborne fraction.

Thank you for catching this inconsistent language. The reviewer is correct that “atmospheric flux ratio” and “emissions ratio” referred to the same figure of merit. As suggested, we have removed the offhanded reference to “emissions ratio” and ensured consistent vocabulary throughout the text.

We have also expanded on the function and purpose of the atmospheric flux ratio and eliminated the comparison to the airborne fraction.

Paragraph four has a couple of hyphens which are problematic. This punctuation occurs several times in the text. It appears that commas or parentheses would be more clear.

We have replaced hyphens with commas and parentheses as appropriate throughout the text.

In the last sentence of paragraph 5 I think we infer that there is a complete linearity between temperature and the CO₂ increase and that a doubling of preindustrial CO₂ would lead to a temperature increase of 3 degrees C.

The reviewer's inference is, to first approximation, correct. *Ceteris paribus*, an equilibrium climate sensitivity (ECS) of 3.0 does imply that a doubling of preindustrial CO₂ would lead to 3°C warming. We have taken this opportunity to expand on the definition of the ECS parameter and its role in climate system feedbacks as well as in the model in the Methods section: (cf. lines 462-468)

A negative feedback loop incorporates heat transfer from the atmosphere and the upper ocean into space via outbound longwave radiation. The magnitude of this feedback, or cooling, is determined by the equilibrium climate sensitivity (ECS), a metric used to characterize the response of the global climate system to a given forcing. ECS is broadly defined as the equilibrium global mean surface temperature change following a doubling of atmospheric CO₂ concentration. In the FeliX model, ECS is nominally set equal to 3.0°C/2×CO₂. In Fig. S5, we plot RAF and ΔT projections based on ECS' = 2.5°C/2×CO₂. This range of values is consistent with C4MIP results⁵ and the latest estimates from IPCC³⁸. □

In the results section, line 1, presumably the IPCC data are found in reference 2. In general the first 2 paragraphs of this section are verbally complex and awkwardly written.

We have thoroughly edited the results section for clarity (cf. lines 73-88).

Page 3, first full paragraph, I think we do not have a shortage of poverty. I do not mean to pick specifically on this sentence but I think it is symptomatic of some of the awkward writing throughout the paper.

The reviewer's point about the general need for precision and accuracy in the text is well taken. In this case, we have changed the specific sentence to: (cf. lines 151-153)

“unnecessary deforestation,” or forest loss resulting from poverty, conflict, poor governance, lack of knowledge, and other causes of failure to optimize land use.

See also the word “standalone” in the figure caption above.

We changed the figure caption to:

“For these variations, energy consumption in 2100 is displayed as columns at right” (cf. Fig. 1)

Figure 1 well illustrates the challenge of presenting a great deal of information in a single picture. The legend is difficult to read but apparently the hydro and nuclear band is in yellow, but it is not clear what the black line of IEA data represents. Presumably the market share and atmospheric CO₂ data at the bottom of each figure are time linked (at 25 year increments?)

The reviewer is correct in observing that we have tried to convey a significant amount of information in Figure 1. However, much of the setup for the results and discussion

comes from comparisons among the four scenarios, so we have tried in the following ways to clarify the figure and its description without fragmenting it:

- We increased the font size in the legend.
- We changed “IEA data” to “Total demand (IEA)”
- Caption:
 - Time series of global primary energy (PE) consumption in the (a) Fossil Fuels, (b) BAU, (c) RE-Low, and (d) RE-High scenarios. Within each scenario, low and high variations on total energy demand (with market share of each fuel held constant) are also modeled. For these variations, energy consumption in 2100 is displayed as columns at right. At bottom, time series of fossil fuel market share (as a percentage of total energy consumption) and atmospheric CO₂ concentration are displayed at quarter-century time intervals. Source of historical data on total primary energy demand: International Energy Agency⁴.

Figure 2 is not adequately described. The colors of so many lines are hard to distinguish, the “ECS” is defined only 1 place in the text, the “CCS25” is defined nowhere. The brown, red, and orange vertical bars seem clear, the blue and green do not. The points with apparent error bars are not described at all.

We have refined the figure and its caption to clarify its contribution to the analysis:

Caption: The atmospheric flux ratio (RAF) is calculated annually as the ratio of net anthropogenic carbon emissions (energy and land use emissions minus anthropogenic sinks) to net carbon sequestration by global plant, soil, and ocean systems. Shaded ranges around the central value of each scenario indicate the effects of $\pm 20\%$ shifts in primary energy demand (cf. Fig. 1). Global surface temperature anomalies projections (ΔT) in 2100 are indicated at right, where each colored bar treats the RE-Low and RE-High scenarios as the endpoints of a continuous range of energy sector decarbonization, plus a fixed rate of CCS. Source of historical values of RAF, indicated by green bars: IPCC³.

Note: ECS is still defined for non-expert readers in context in the sensitivity analysis and methods sections (cf. above)

On page 4 the paragraph on “primary energy demand” is poorly connected to the rest of the text and poorly explained. Or maybe it is just in the wrong place in the text.

We’ve rolled this paragraph into the sections on each scenario’s results and into the scenario sensitivity analysis.

Figure 3 has challenges similar to those in figure 2. The little dots with error bars are not explained, it takes some digging to understand why there appear to be 2 lines for the historical data, the legend does not appear until Figure 3b, and one really needs some expertise in the field to understand the appearance of the RCP lines. My goodness there is a lot of information in Figure 3. This is not for the faint of heart.

In response to this comment, we split the figure in question into 3 separate figures: energy emissions (now Fig. 3), LULUCF emissions (Fig. 4) and carbon flux through the atmosphere, ocean, and land sinks (Fig. 5).

As mentioned above, we have also eliminated two of the six CCS scenarios, which should make the plots more accessible.

The appropriateness of showing the RCPs is a question on which we recognize opinions differ. We believe they are useful for experts to contextualize FeliX results in the larger field of research. Further, consistency between FeliX scenarios and RCPs across multiple indicators (i.e. emissions, radiative forcing, and warming) should increase in expert readers confidence in the soundness of the bio-geophysical mechanics underlying our scenarios.

For the benefit of both expert and lay readers, we have clarified in all figures in which they appear that the RCPs are independently-derived benchmark emissions pathways from the IPCC, and shown for comparison. We have also added the following to the discussion (cf. Lines 257-268):

Our analysis extends beyond the IPCC benchmarks in that the Representative Concentration Pathways are not fully-integrated scenarios of socio-economic, energy and LULUCF emission, and climate projections, but rather internally-consistent pathways of atmospheric greenhouse gas concentrations. Indeed, the RCPs are intended as inputs to coupled emissions-carbon cycle-climate models like FeliX. Thus, although the scenarios in this analysis are not exogenously linked with specific RCPs, our reasons for drawing attention to points of comparison between the two sets are twofold: first, comparisons across multiple indicators (i.e. emissions, radiative forcing, and warming) serve to establish the internal consistency of the FeliX model. Second, comparisons between FeliX scenarios and the RCPs serve to link detailed, particular energy and LULUCF transitions with "representative" pathways, and mark progress toward the COP target.

On page 6 the first partial paragraph has text "this mitigation can up to a certain point be achieved..." which is very unclear. I thought that all of this was already rolled into the scenarios. So what are we saying here?

These comments were intended to anticipate the criticism of CCS skeptics, but we agree they are unhelpful for understanding the scenarios he have devised, so we have eliminated the passage in question, and replaced it with the following (cf. lines 293-299):

We also note that, when it is achieved in the 1/2CCS and CCS emissions pathways and in similar analyses, full decarbonization relies on the coupling of CCS technology with bioenergy production (BECCS), a carbon-negative process. If BECCS is ultimately found to be unfeasible, uneconomical, or unacceptably burdensome on ecosystems, then alternative negative emissions technologies (eg. direct air capture) will need to be substituted. Absent these failsafes, fossil fuels will need to be phased out completely, and well before 2100.

Page 7, the first sentence under the heading "net negative.." uses the word "more", more

than what? The same paragraph gets to the idea shown in Figure 3c that “negative atmospheric carbon flux ultimately leads to a net outflow of carbon from plant, soil, and ocean reservoirs...” This becomes sufficiently important in the conclusions that I suggest it needs some explanation of the physics involved. My understanding is that net flows between the ocean and atmosphere depend on the concentration gradient and thus only indirectly on anthropogenic emissions.

We include the following explanation of this effect in the Methods section of the manuscript: (cf. lines 421-433)

The simplest physical explanation for this effect is that rapid decarbonization reverses the sign, or direction, of the chemical coupling between the atmosphere, ocean, and land sink. This coupling is responsible for increasing the net carbon flux into natural sinks as emissions rise, so it should also be expected to cause a net carbon flow out of these sinks if net-negative anthropogenic emissions are achieved and atmospheric carbon concentration begins to drop.

From another perspective, we could invoke Le Châtelier’s principle to predict that the earth system will act to “resist” change. In deep decarbonization scenarios, the “change” is the reduction in atmospheric carbon concentration resulting from net-negative anthropogenic emissions, and the natural response is the net flow of carbon out of the land and ocean sinks.

In reference 7 I suspect that author Van Vuuren would like to have the Van with his name here.

Thank you for this impressive and very constructive level of attention to detail. We are indebted to the reviewers’ work, as we believe the paper is much better for their work.

I did not read all of the way through the supplemental material.

As requested by the editor, we have moved as much of the SM as possible in to the Methods section of the manuscript in order to broaden the scope and accessibility of the analysis.

My overall reaction is that for such a complex paper to be accessible to a wide audience it needs a smoothly flowing text and meticulous attention to detail.

Reviewer #2 (Remarks to the Author):

This paper analyzes a series of alternative greenhouse gas emission and CO₂ removal pathways to limit global warming. It focusses on one of the goals agreed by the Conferences of Parties (COP) in the recent Climate Conference in Paris in December 2015: "... to achieve a balance between anthropogenic emissions by sources and removals by sinks of greenhouse gases in the second half of this century." The goal was introduced in support of the primary climate goal that for many years has been the central subject of the COP conferences: the limitation of global warming to maximally 2deg C. The relation between the recent Paris balanced-source-sink goal and the long-standing COP 2 deg C global warming goal is an interesting scientific question, with considerable practical political implications. It can be expected to appeal strongly to the general reader. Unfortunately, however, the inter-relationship is not discussed in detail in the paper, but referred to only marginally. There is also no reference to other attempts to translate the 2 deg C goal into practical political implications, such as the limitation this imposes on the maximal global per capita CO₂ emissions.

This point is very well taken, and we thank the reviewer for pointing out the incomplete linkage here. In general, we feel that FeliX adds value to the field of earth system and impact modeling because we do not treat the model as a "black box," nor do we require readers to do so. The model should be accessible, both in terms of the scrutability of its components and its availability for download.

We have worked in this draft of the manuscript to present more clearly the Results and Discussion sections with an eye toward better establishing the inter-relationship between the timing of net-zero emissions and 2degC. In addition, we have expanded the Methods section of the manuscript considerably in order to clarify the physical and methodological basis for the global carbon cycle. This includes expanded discussions of equilibrium climate sensitivity, chemical and temperature feedbacks, alternative non-CO₂ emissions pathways. Finally, we have expanded considerably the error analysis in order to show the dependence of scenario results on fundamental model parameters.

These shortcomings reflect a general tendency of the authors to follow their own path, without detailed reference to other work. Thus the new set of emission pathways analyzed by the authors, although of considerable interest in their own, are not tied in to the extensive literature that already exists on climate mitigation computations. There are indeed references (7,8) to the "representative concentration pathways" project, a new major international effort to coordinate mitigation analyses, and figures 3 and 4 actually include RCP data as "benchmarks". But there is no explanation this reviewer could discover, either in the figure or elsewhere, of how the RCP data relate to present paper (apart from a cryptic comment "... resulting in an emissions pathway that is comparable to RCP 4.5...." in the last paragraph before the Discussion section).

We have made a significant effort in this revision to clarify the relationship of FeliX scenario to the RCPs and, by extension, to the large bodies of research and literature they represent (cf. lines 257-268):

Our analysis extends beyond the IPCC benchmarks in that the Representative Concentration Pathways are not fully-integrated scenarios of socio-economic, energy and LULUCF emission, and climate projections, but rather internally-consistent pathways of atmospheric greenhouse gas concentrations. Indeed, the

RCPs are intended as inputs to coupled emissions-carbon cycle-climate models like FeliX. Thus, although the scenarios in this analysis are not exogenously linked with specific RCPs, our reasons for drawing attention to points of comparison between the two sets are twofold: first, comparisons across multiple indicators (i.e. emissions, radiative forcing, and warming) serve to establish the internal consistency of the FeliX model. Second, comparisons between FeliX scenarios and the RCPs serve to link detailed, particular energy and LULUCF transitions with ``representative'' pathways, and mark progress toward the COP target.

A minor comment, but in the same vein: it would be helpful for the general reader to have better explained why the authors prefer to introduce as the basic variable of their paper the atmospheric flux ratio $RAF = x/y$, where $x =$ net anthropogenic CO₂ emissions and $y =$ net CO₂ uptake by natural sinks, rather than the familiar airborne fraction $AF = (x - y)/x$. RAF is clearly not "the airborne fraction inverse" as stated by the authors (second paragraph). The rest of the paragraph nonetheless clearly explains the useful information content of RAF. But the first two paragraphs of the following section "Results" are again less clear. The relation between x and y is better described, not in terms of "feedback", but by the fact that while x is an externally prescribed variable at any given instant, y represents an integral system response dependent on the past history of the system forcing x , and will thus always adjust in time to any change in x . There is no feedback of y on x .

We thank the reviewer for this comment. We have included in this round of revisions a much clearer articulation of RAF and the ways we see it as useful for interpreting the FeliX scenarios (cf. Lines 30-62):

Before proceeding, we offer a few remarks about RAF and its aptness as a figure of merit. First, we acknowledge that the roles of the ocean and land sinks are often left implicit in carbon budget analyses, and the text of the Paris Agreement itself leaves ambiguous the definition of ``net zero emissions'' insofar as it does not specify whether sinks need be directly anthropogenic (i.e., deliberately created as a result of mitigation policy) or natural (i.e., caused inadvertently by the human perturbation of the carbon cycle). We have included natural sinks in this analysis in order to present a complete accounting of global carbon flows, and because the carbon flux from the atmosphere into other parts of the Earth system (i.e. biomass, soil, and the oceans) will be essential to the magnitude and timing of economic and technological interventions necessary to realize the COP targets.

Second, carbon emissions pathways impact natural sinks through carbon and temperature couplings in the FeliX model, but the sinks do not feed back into anthropogenic emissions from either the energy or LULUCF sectors. While some mechanisms could complete this feedback loop (eg. carbon credits in a hypothetical global cap and trade market), they are not included in this analysis. For this reason, RAF separates the external, prescribed drivers of each scenario (i.e. anthropogenic emissions) from the internal, dynamic responses (i.e. natural sinks).

Apart from these general comments - which can hopefully be responded to without major modifications - the paper represents a very useful and readable analysis of the impact of

various feasible mitigation strategies in response to the secondary COP 21 goal of neutralizing carbon emissions by the end of this century. The figures are well organized and clear, and the impact of different combinations of the various levels of renewable-energy and carbon-capture-and-sequestration technologies well presented.

Thank you for this assessment. In response to other reviewers' comments, we have edited several of the figures and rearranged them in the paper, and we are confident that readers will find them much more accessible in this draft.

Of interest, in particular, is the critical role of carbon capture and sequestration, which needs to be introduced on a major scale in addition to renewable energy to meet the COP 21 goals. This reviewer would have welcomed a more detailed discussion of the political implications of the role of CCS, which is still in its infancy regarding large scale implementation compared to that of renewable energy, which, despite its present low level of global implementation, is technically more advanced. However, it is understandable that the authors may prefer to leave a more detailed discussion of this important question to follow-up publications.

Indeed, we have planned a follow up analysis that looks more closely at the relationship between the fossil fuel phase-out tail and the availability of BECCS and other negative emissions technologies.

However, the reviewers point is well-taken. We have highlighted the reliance of individual scenarios on CCS in the Results section (cf. lines 188-224) and included the following comments about the role that CCS—and, more specifically, BECCS—is likely to play in decarbonizing pathways: (cf. lines 293-310):

We also note that, when it is achieved in the 1/2CCS and CCS emissions pathways and in similar analyses, full decarbonization relies on the coupling of CCS technology with bioenergy production (BECCS), a carbon-negative process. If BECCS is ultimately found to be unfeasible, uneconomical, or unacceptably burdensome on ecosystems, then alternative negative emissions technologies (eg. direct air capture) will need to be substituted. Absent these failsafes, fossil fuels will need to be phased out completely, and well before 2100.

Conversely, if the decarbonization of the energy and LULUCF sectors does not proceed apace or if nutrient supply limits land sink carbon uptake, then carbon sequestration technology will need to be employed as quickly and as broadly as possible to meet the COP targets. Depending on the economic feasibility of carbon feedstocks for chemical production and chemical energy storage, permanent sequestration of up to an exagram of carbon (1,000 PgC) may be required. This may be technically possible, as global permanent storage capacity has been estimated to range from 135~PgC to as high as 2,700~PgC. However, large scale remains a distant reality, given a current global annual geosequestration rate of only 7.3~MtC. In addition, it is not yet clear how much the energy and infrastructure costs of CCS will benefit from economies of scale.

In summary, I recommend publication with minor modifications.

Klaus Hasselmann, em.Dir., Max Planck Institute for Meteorology, Hamburg

Reviewer #3 (Remarks to the Author):

General comments

This manuscript describes an analysis of the effects of a variety of future scenarios of emissions, land use change, and technology change, oriented in particular around the ratio of anthropogenic emissions to natural sinks. This is potentially interesting, especially given that the authors are using a relatively new model and non-standard (i.e. not RCP) scenarios. The ms is well written, easy to read, and generally interesting.

There are a number of issues that concern me here, however. First, I'm unconvinced by the utility of Raf, the authors' metric of choice; it seems confusing and difficult to interpret, or at least vulnerable to misinterpretation. See specific comment below.

<Addressed below>

Second, FeliX is not well known in the integrated assessment world (NB I am not an integrated assessment modeler, but am reasonably familiar with the field), and a bit more information in the text about its design and limitations would be useful. For example, as far as I can tell its uses relationships, based on historical information, between key parameters that are then forecast forward. It doesn't seem to have any technological detail, process detail, geographic regions, nor any physical constraints (e.g., land area for food, oil resources). It would be good to discuss this a bit, acknowledging limitations and strengths--i.e. why you chose this model design.

The model does have many of these features, and we have expanded the Methods section of the manuscript to make the structure of the model clearer:

(cf. lines 349-353)

While there are significant uncertainties associated with projecting energy consumption several decades into the future, primary energy profiles are not FeliX model results for the purposes of this analysis. Rather, they are definitional of their respective scenarios, and are intended to illustrate correspondences between RAF and energy sector transformations of various magnitudes.

(cf. lines 375-388)

Land in the FeliX model is distributed among four mutually exclusive and collectively exhaustive categories: agricultural, forest, urban/industrial, and "other". Each category is calibrated to FAOSTAT data on a global level (available for 1961-2010 for agricultural and 1990-2012 for forest and other land). Though not on a geographically explicit basis, land can be repurposed--most notably, due to changes in demand for agricultural land.

Land categorized as "agricultural" is subdivided into arable land, permanent crops, and permanent meadows and pastures. Arable land and permanent crops can be used to produce food, feed, or energy crops, while permanent meadows and pastures are used only for feed production. The BAU scenario is calibrated to historical data available on FAO. Crop and livestock yields are modeled endogenously as a function of input-neutral technological advancement, land management practices (i.e. the expansion of high-input agriculture), water availability, pollution (including atmospheric carbon fertilization), and climate change.

In addition, the Felix model is documented in its entirety on the model webpage <www.felixmodel.org>, where it is also freely available for download.

Third, some of the physical parameters or characteristics of the model are not well documented and seem questionable (see specific comments below). In particular, Felix's assumption of preindustrial NPP is 50% higher than most other estimates, if I understand correctly, which raises doubts about the fidelity of the model's entire C cycle. This needs to be clarified/justified/explained much better.

<Addressed below>

Finally, the discussion spends too much time restating results, and doesn't really compare these results to previous, similar studies, or provide much in the way of context. As a result the novelty of this work is not clear.

Thank you for this useful comment. We have revamped the Discussion section thoroughly to avoid restatement of results, better contextualize analysis results with the RCPs; and to make clearer the value-add of the results.

In summary, there are many points of interest here, but the current ms needs extensive revisions to better explain and justify the model choice, parameters, Raf ratio, and better contextualize the results with respect to previous work in this field.

Specific comments

1. *In the future please number lines—makes reviewing much easier*

We have reformatted the PDF into a single column and added line numbering. For ease of dialogue going forward, we have added line numbers to specific comments as appropriate.

2. *Page 1: why is defining an "Raf" ratio necessary, useful, or appropriate? After all, the effects of atmospheric CO₂ are determined by its absolute level in the atmosphere, not the ratio of inflows to outflows; and a ratio is also subject to distortion at, e.g., small flux levels (as it's nonlinear with respect to flux size), which makes interpreting it confusing. So I think that this needs better, clearer justification*

Thanks for this insightful comment. We have taken this revision as an opportunity to reconsider and, ultimately, better articulate the utility of R_{AF} in this analysis. There are two primary considerations: first, the ratio allows us to characterize the state of the global carbon cycle with a single figure (admittedly broadly and imperfectly). Second, the ratio as constructed allows us to identify 3 qualitative emissions states: (cf. lines 30-62)

The concept of a carbon budget involves multiple dynamic, interrelated components of the global carbon cycle, and can be defined in a number of ways. As a figure of merit for this analysis, we define an atmospheric flux ratio (R_{AF}) as the ratio of net CO₂ emissions (anthropogenic sources minus artificial sinks) to net CO₂ uptake by natural sinks (i.e. plant, soil, and ocean systems).

The atmospheric flux ratio characterizes the instantaneous state of the global carbon system relative to the COP targets. Atmospheric flux ratios greater than

unity ($R_{AF} > 1$) indicate increasing atmospheric carbon concentrations, associated radiative forcing, and temperatures. Ratio values between zero and unity ($0 < R_{AF} < 1$) indicate net negative atmospheric carbon flux due to net ocean and land sink uptake, an important milestone on the path to climate stabilization. Finally, values below zero ($R_{AF} < 0$) indicate net negative anthropogenic emissions, or the achievement of the COP target for carbon emissions. We calculate recent historical values of R_{AF} on the basis of data from the Intergovernmental Panel on Climate Change (IPCC), and use the model to project R_{AF} through 2100 for all scenarios.

Before proceeding, we offer a few remarks about R_{AF} and its aptness as a figure of merit. First, we acknowledge that the roles of the ocean and land sinks are often left implicit in carbon budget analyses, and the text of the Paris Agreement itself leaves ambiguous the definition of "net zero emissions" insofar as it does not specify whether sinks need be directly anthropogenic (i.e., deliberately created as a result of mitigation policy) or natural (i.e., caused inadvertently by the human perturbation of the carbon cycle). We have included natural sinks in this analysis in order to present a complete accounting of global carbon flows, and because the carbon flux from the atmosphere into other parts of the Earth system (i.e. biomass, soil, and the oceans) will be essential to the magnitude and timing of economic and technological interventions necessary to realize the COP targets.

Second, carbon emissions pathways impact natural sinks through carbon and temperature couplings in the Felix model, but the sinks do not feed back into anthropogenic emissions from either the energy or LULUCF sectors. While some mechanisms could complete this feedback loop (eg. carbon credits in a hypothetical global cap and trade market), they are not included in this analysis. For this reason, R_{AF} separates the external, prescribed drivers of each scenario (i.e. anthropogenic emissions) from the internal, dynamic responses (i.e. natural sinks).

3. P. 4: "and climate" not "an climate"

Thank you for catching this typo, which we have corrected.

4. P. 9: preindustrial terrestrial NPP of 85 Pg C/yr? That's really high—most ESMs, remote-sensing analyses, etc., put current NPP much lower. For example:

- the CASA model simulates ~48
- Nemani et al. estimated ~54 (<http://science.sciencemag.org/content/300/5625/1560>);
- FLUXNET syntheses, etc., all produce numbers in the 46-56 range as far as I'm aware. Where did this 85 come from?

The nominal value of 85 PgC/yr came from the C-ROADS model, on which the structure and calibration of the Felix model's carbon cycle are based. This value was reaffirmed in the *C-ROADS Simulator Reference Guide* as recently as 2015.

However, the reviewer's point about the Felix model's NPP being higher than independent estimates is well taken. We reran the BAU scenario with the initial NPP value set to 51 PgC/yr to estimate the effect of this shift, with the following results:

	Units	2015		2100	
		BAU (NPP _I =85.2)	BAU* (NPP _I =51.0)	BAU (NPP _I =85.2)	BAU* (NPP _I =51.0)
Annual flux Atm. to Land Sink	PgC yr ⁻¹	2.71	1.77	2.59	1.61
Cum. flux Atm. to Land Sink	PgC	123	81.4	380	245
Annual flux Atm. to Ocean	PgC yr ⁻¹	2.71	2.90	3.18	3.30
Cum. flux Atm. to Ocean	PgC	147	159	417	443
Atm. CO ₂	ppm	408	423	712	763
Temperature Anomaly	°C	1.16	1.25	3.23	3.44

Annual carbon flux into the land sink in BAU* is reduced by 35% in 2015 and by 38% in 2100 relative to BAU results in the same years. Cumulative uptake by the land sink declines by 34% from 1900 through 2015 and by 36% through 2100. These results are at the lower end of IPCC land sink flux estimates.

Annual carbon flux into the ocean in BAU* increases by 7% in 2015 and by 4% in 2100 relative to BAU results in the same years. Cumulative uptake by the ocean increases by 8% from 1900 through 2015 and by 6% through 2100. These results are modestly closer to the central IPCC value than the BAU scenario, though both scenarios are well within the IPCC range.

Atmospheric CO₂ concentration rises 4% in 2015 and 7% in 2100. Agreement with historical data on atmospheric carbon content is worse in the BAU* scenario than in BAU. The global surface temperature anomaly in BAU* rises by 0.1°C in 2015 and by 0.2°C in 2100 relative to BAU.

Overall, the effect of the NPP shift on both atmospheric carbon and the global temperature anomaly is roughly equal to the difference between the low and high energy demand variations on the BAU scenario.

We have included in this revision of the analysis a table that documents the effect of NPP and other essential model parameters on C_{atm} (cf. Supplementary Table 2)

5. P. 9: And where did these turnover times (10.6 years for biomass, 27.8 years for soil) come from? Are they consistent with the most recent, best estimate I'm aware of, Carvalhais et al. 2014 (10.1038/nature13731)?

The turnover times for biomass and soil also come from the C-ROADS model. In this case, these figures are quite consistent with the central value of the Carvalhais analysis ($\tau_{2015}^{Carvalhais} = 23$ years):

$$\tau_{2015}^{Felix} = \frac{C_{total}}{gross\ flux\ out} \Big|_{2015} = 22.20\ \text{years}$$

The average turnover time declines modestly to 21.8 years by 2100. This is due to climate feedback to humus residence time.

The Carvalhais analysis assigns errors of +7 years and -4 years to their central value. We have adopted the same range to propagate uncertainty on this parameter to C_{atm} (cf. Supplementary Table 2)

6. P. 10: “emerging results” from CMIP5? Isn’t CMIP5 pretty well ‘done’ at this point?

Thank you for catching this misstatement. We have expanded the discussion of the relationship of the C4MIP and CMIP5 projects to the FeliX model and results, incorporating comments 4-7 from this review.

7. P. 10: What exactly does “gain” in table 2 mean?

Gain is a parameter introduced in *Climate Sensitivity: Analysis of Feedback Mechanisms* by Hansen, et. al. (1984). It describes the contribution of different mechanisms to total climate change. In the context of FeliX and C4MIP, model gain (g) describes the ratio of temperature change due to land and ocean sink feedbacks to total temperature change.

We have included the Hansen paper in the references (39) and added the following explanation to the text: (cf. lines 527-536).

“Overall gain (g) of the climate system, which quantifies the ratio of temperature change due to these feedback loops to total temperature change, is shown in the column at far right”

Reviewer #4 (Remarks to the Author):

A. The authors propose a new assessment of historic and future global atmospheric emissions of carbon using the FeliX model. The model takes into account economic and natural parameters to compute the annual stocks and flows (emissions and sinks) of carbon.

B. This study is original in the sense that it considers the impact of human activities and natural sinks in the global carbon cycle, and relies on the ratio of the first over the latter (the atmospheric flux ratio R_{AF}) to quantify pathways for various policy options. This indicator, as well as carbon emissions and temperature anomalies, are computed until 2100.

C. Most data can be deemed robust as it comes from international bodies such as IEA, and as such there is a consensus on its validity. Similarly, the scenarios tested are consistent with each other.

D. Uncertainty analysis is crucial in this type of exercise, especially for such long-term storylines. As far as I can see, scenarios and their variants account for enough uncertainty for the results to be considered robust. All figures are reported with confidence intervals, and sensitivity analyses and error analyses are performed for all scenarios. Clearly, including feedbacks in climate modelling introduces major uncertainty, but the authors have made serious efforts to understand and quantify it.

We have further expanded the uncertainty analysis, looking at the sensitivity of atmospheric carbon concentrations in the BAU scenario to a wide range of model parameters (cf. Supplementary Table 2).

E. The study is robust overall, and can be seen as a valuable addition to existing knowledge on the contribution of anthropogenic carbon emissions to the carbon cycle. The results are surprisingly optimistic, as the consensus tends to be that 2°C warming objectives are becoming unattainable, which makes the conclusions of this study very interesting. Of course, increasing decarbonization of the anthroposphere is necessary, but gives hope concerning the COP emissions target.

F. The work looks very much final, and I do not have much to say about improvements. I would recommend the publication of this work without significant change to the manuscript. More scenarios could of course always be included, but keeping the sources limited to the IEA and IPCC provides an overall consistency and robustness - and I believe that FeliX is flexible enough to test other scenarios easily.

G. The list of references may remain unmodified.

H. The manuscript is clear and provides factual information throughout the various sections, I could not find a claim that came unsubstantiated.

REVIEWERS' COMMENTS:

Reviewer #1 (Remarks to the Author):

Having read an earlier draft of this paper I can offer the opinion that the figures are vastly improved and that the text flows much more smoothly. I think that it has some nice text on the role of natural sinks (notably lines 251-254). It is still a complex paper that covers a lot of ground, but I think now it is accessible to a more general reader. The methods section will be for the hard core readers. I have 2 quick comments.

1.) line 67 mentions RCP without reference or explanation. By the end of the paper there is probably enough explanatory text on the RCP scenarios, but a little something out here in front would be very helpful for the general reader.

2.) The impacts and implications of CCS are so significant that I think the paragraph beginning on line 195 should say a little more about the implementation rate. Lines 362-368 would help up here in front but the "ramps up rapidly from 2020 through 2040" text with supplementary figure 1a puts it really in perspective.

And just to assure you that I read the whole thing, please note that in the caption to supplemental figure 4, last line, the verb "is" should be "are". There are also two typos in the last 4 lines of the abstract: make it "to be reduced" and change the "of" to "or".

It is an interesting and useful paper with a novel and creative perspective.

Reviewer #2 (Remarks to the Author):

Review of 2nd revised manuscript:

Pathways for balancing CO₂ emissions and sinks. Brian Walsh et al.

The authors have made a strong constructive effort to improve their paper in response to the reviewers' comments. It now reads much better.

In particular, my concerns that the paper failed to adequately explain the relation of their scenario computations to the scenario computations of other groups, particularly the extensive set of Representative Concentration Pathways produced by a large international collaboration, have been largely responded to. However, it would have been better for the authors to have explained this relation up front, in the introduction, rather than later in a brief mention in lines 114-117 and then in more detail in the discussion at the end, lines 257-268.

A few minor comments:

Lines 35-36. The atmospheric flux ratio characterizes the instantaneous state [no: rate of change] of the global carbon system relative to the COP targets [no, is not "relative to COP targets" but is simply important to understand to plan policies addressing the COP targets]

Lines 41-42. ", or the achievement of the COP target for carbon emissions". Meant is presumably: "an important and probably necessary step for the achievement"

Figure 1, small-letter comment right of the figure: "ECS= 3 degC/2xCO₂". I thought the authors had somewhere stated that their ECS value was about 2.5 degC/2xCO₂.

Line 80: ".the distribution of carbon [uptake} among..."

Line 319: typo: an "as" too many

I recommend publication, with the minor revisions suggested above.

Reviewer #3 (Remarks to the Author):

The authors have done a nice job addressing my previous concerns, in particular with respect to the atmospheric flux ratio, as well as details of the Felix models and its inputs. The manuscript is now clearer and stronger throughout, and I have no further concerns.

REVIEWERS' COMMENTS:

Reviewer #1 (Remarks to the Author):

Having read an earlier draft of this paper I can offer the opinion that the figures are vastly improved and that the text flows much more smoothly. I think that it has some nice text on the role of natural sinks (notably lines 251-254). It is still a complex paper that covers a lot of ground, but I think now it is accessible to a more general reader. The methods section will be for the hard core readers. I have 2 quick comments.

1.) line 67 mentions RCP without reference or explanation. By the end of the paper there is probably enough explanatory text on the RCP scenarios, but a little something out here in front would be very helpful for the general reader.

Author response:

Both reviewer #1 and #2 noted that the RCPs are inadequately explained the first time they are mentioned in the text. We have addressed this issue with a brief definition of the RCPs in lines 68-70:

Emissions pathways and associated radiative forcing for non-CO₂ greenhouse gases (i.e. CH₄, N₂O, HFC, and "others") are assumed to follow Representative Concentration Pathway (RCP) 4.5, one of the IPCC's four benchmark pathways for atmospheric greenhouse gas concentrations, through 2100.^{4,5}

2.) The impacts and implications of CCS are so significant that I think the paragraph beginning on line 195 should say a little more about the implementation rate. Lines 362-368 would help up here in front but the "ramps up rapidly from 2020 through 2040" text with supplementary figure 1a puts it really in perspective.

Author response:

We agree with the reviewer that the expansion rate of CCS should be made clear—in particular, because one point of the exercise is to show how ambitious CCS expansion will need to be in order to become the panacea some imagine it will be. To this end, we added the following lines to the results section (206-214):

In order to achieve these capture rates while keeping up with primary energy demand growth, the 1/2CCS and CCS pathways assume geometric growth (30% yr⁻¹ and 34% yr⁻¹, respectively, from 2016 through 2040) in the amount of new carbon sequestered each year. Relative to the actual current value (7.3 MtC yr⁻¹)¹⁴, annual carbon capture expands by a factor of 630—720 by 2040 in 1/2CCS scenarios, and by a factor of 1310—1490 by 2040 in the CCS scenarios. From 2040 through the end of the century, CCS capacity expands an additional 10—40% in both pathways to account for rising marginal costs of additional efficiency improvements and infrastructural expansion (cf. Supplementary Fig. 1a).

And the following comment to the discussion section (319-322):

On the other hand, this low baseline makes sustained 30—35% year-over-year growth in the geosequestration rate a technologically realistic target, even before economies of scale begin to lower the energy and infrastructure costs of CCS.

3.) And just to assure you that I read the whole thing, please note that in the caption to supplemental figure 4, last line, the verb "is" should be "are". There are also two typos in the last 4 lines of the abstract: make it "to be reduced" and change the "of" to "or".

Author response:

Thanks very much for your time and close attention. We have addressed these typos.

It is an interesting and useful paper with a novel and creative perspective.

Reviewer #2 (Remarks to the Author):

Review of 2nd revised manuscript:

Pathways for balancing CO₂ emissions and sinks. Brian Walsh et al.

The authors have made a strong constructive effort to improve their paper in response to the reviewers' comments. It now reads much better.

1.) In particular, my concerns that the paper failed to adequately explain the relation of their scenario computations to the scenario computations of other groups, particularly the extensive set of Representative Concentration Pathways produced by a large international collaboration, have been largely responded to. However, it would have been better for the authors to have explained this relation up front, in the introduction, rather than later in a brief mention in lines 114-117 and then in more detail in the discussion at the end, lines 257-268.

Author response:

We refer to our response to the first reviewer's first comment, above (cf. lines 68-70).

2.) A few minor comments:

Lines 35-36. The atmospheric flux ratio characterizes the instantaneous state [no: rate of change] of the global carbon system relative to the COP targets [no, is not "relative to COP targets" but is simply important to understand to plan policies addressing the COP targets]

Author response:

We have edited the sentence in question to state more directly (cf. line 36):

The atmospheric flux ratio characterizes annual changes in the atmospheric carbon burden.

3.) Lines 41-42. “, or the achievement of the COP target for carbon emissions”. Meant is presumably: “an important and probably necessary step for the achievement”

Author response:

By construction, negative values of R_{AF} are identical to the COP emissions target. We have edited the sentence in question for clarity (cf. line 41-43):

Finally, values below zero ($R_{AF} < 0$) indicate net negative anthropogenic emissions—that is, the achievement of the COP carbon emissions target.

4.) Figure 1, small-letter comment right of the figure: “ECS= 3 degC/2xCO₂”. I thought the authors had somewhere stated that their ECS value was about 2.5 degC/2xCO₂.

Author response:

The nominal value for ECS, used to derive all results, is 3.0 degC/2xCO₂. We have double checked that all figures note this value, and confirmed that the alternative value of 2.5 is limited to the SI.

**5.) Line 80: “..the distribution of carbon [uptake] among...”
Line 319: typo: an “as” too many**

Author response:

Thank you for your close read. We have corrected these typos & proofread for additional errors.

I recommend publication, with the minor revisions suggested above.

Reviewer #3 (Remarks to the Author):

The authors have done a nice job addressing my previous concerns, in particular with respect to the atmospheric flux ratio, as well as details of the Felix models and its inputs. The manuscript is now clearer and stronger throughout, and I have no further concerns.

Author response:

We thank and are indebted to the reviewers for their generous investment of time and energy, which have resulted in a much stronger analysis.